# Dickkopf1 induces enteric neurogenesis and gliogenesis in vitro if apoptosis is evaded

Melanie Scharr [1], Simon Scherer[2], Bernhard Hirt[1] & Peter H. Neckel [1]✉

Neurogenesis in the postnatal enteric nervous system (ENS) is controversially discussed. Yet, deciphering the regenerative potential of the ENS is essential for our understanding and therapy of human enteric neuropathies. Dickkopf1 (DKK1) is a Wnt-antagonist and involved in the homeostasis of various tissues. We hypothesize that DKK1 could function as a negative regulator on the proliferation of ENS-progenitors in the postnatal gut of mice and human infants. Here, we provide evidence that DKK1 is expressed in the murine and human ENS. If applied to ENS-progenitors in vitro, DKK1 leads to an increased proliferation, however, followed by extensive apoptosis. Yet, once we block apoptosis, DKK1-stimulation markedly increases enteric neurogenesis in murine and human ENS-progenitors. Thus, DKK1 is a strong, ambivalent regulator of the ENS-progenitor cell pool in mice and humans. These results are fundamental steps to reshaping our understanding of the homeostasis of the ENS in health and disease.

[1] Institute of Clinical Anatomy and Cell Analysis, University of Tübingen, Tübingen, Germany. [2] Department of Pediatric Surgery and Urology, University Children's Hospital Tübingen, Tübingen, Germany. ✉email: peter.neckel@uni-tuebingen.de

The enteric nervous system (ENS) represents the intrinsic innervation of the gastrointestinal (GI) tract acting partly autonomously and independent of central nervous system processing[1]. ENS reflex pathways control intestinal motility, local blood flow, transmucosal movement of fluids, or secretion of gut hormones[2,3].

Despite these vital functions, the capability of the ENS to adapt to the changing environment of the gut throughout postnatal life has hardly been investigated. Thus, little is known on the regulation of tissue homeostasis of the ENS, both in vivo and in vitro. While enteric neurogenesis in the mature ENS is controversially discussed and is arguably a rare event in vivo[4–9], many studies demonstrated the isolation, expansion, and differentiation of proliferative cells derived from the postnatal ENS of various rodent species and human patients, in vitro[10–14]. This intriguing discrepancy between the proliferative potential of ENS-progenitor cells in vivo and in vitro is likely due to different microenvironments, in particular the presence of pro-proliferative morphogens and matrix components[15]. These regulatory factors of the ENS-progenitor niche drew increasing attention over the last years. Thus, a recent investigation by Stavely et al. showed that enteric mesenchymal cells (EMCs) shape the fate of postnatal ENS-progenitor cells by the paracrine secretion of morphogens. In particular, they found that EMCs express several ligands of the Wnt-signaling pathway with corresponding receptors expressed on ENS-progenitor cells[16].

In this cohesion, the Wnt-signaling pathway plays an essential role in the regulation of proliferation and differentiation of neural stem and progenitor cells[17,18], and has been studied extensively as a key regulator of adult stem cell homeostasis in different metazoan tissues[19]. With respect to the ENS, Wnt-signaling induces and specifies neural crest formation and is involved in regulating enteric neural cell migration, guidance, and growth of enteric neuronal projections in the developing gut[20]. Our group and others have shown that canonical Wnt-signaling increases the proliferative capacity of postnatal ENS-progenitors from rodent models and human infants in vitro and successively leads to a higher yield of newly generated neurons[21,22]. Due to its pivotal role in regulating proliferation and differentiation in various organs, Wnt-signaling is tightly regulated. A prominent group of Wnt/β-catenin-antagonists, are Dickkopf proteins, which are an evolutionary conserved gene family of four glycoproteins (DKK1-4)[23]. The most extensively studied member, DKK1, was initially described as a critical operator in head induction in Xenopus[24]. Differential expression patterns in various neural and mesenchymal tissues[23,25,26] suggests that DKK1 is involved in controlling morphogenesis in diverse tissue types in vertebrates[23]. Recent evidence emerged, that DKK1 is involved in a variety of cellular/context-dependent functions, such as regulating cell proliferation[27] and differentiation processes[28–30], cell survival and programmed cell death[31]. Yet, few data on the function of DKK1 in regulating the ENS-progenitor cell pool is available and thus, the role of DKK proteins in the ENS remains elusive. However, previous data of our group suggest the expression of Dkk-mRNAs in proliferating enterospheres derived from postnatal mice[32].

As Wnt/β-catenin-signaling was reported to increase proliferation in ENS-progenitor cells eventually leading to a higher number of neurons, we hypothesized that the Wnt-antagonist DKK1 functions as a contra-proliferative cue on postnatally derived ENS-progenitors from mice and men in vitro. Our findings suggest, that DKK1 is an important regulatory factor for orchestrating the number of proliferating ENS-progenitors in vitro; however, surprisingly not as initially expected by our working hypothesis.

## Results

**In vivo expression of DKK1 in mouse and human intestine.** To evaluate, the role of DKK1 in the postnatal gut, we first analysed the expression pattern of DKK-ligands/receptors in the small and large intestine of mice and humans with in situ hybridisation and immunhistochemistry. On the mRNA level, Dickkopf-ligands 1-4 (*Dkk1, Dkk2, Dkk3, Dkk4*), and the relevant receptors Kremen 1-2 (*Krm1, Krm2*) and Low-density lipoprotein receptor-related protein 5 and 6 (*Lrp5* and *Lrp6*) were detected within submucosal (Supplementary Fig. 1a) and myenteric plexus of murine small (Fig. 1a) and large intestine (Fig. 1b). In addition, *Dkk*-ligands and corresponding receptors are expressed with different intensities within the epithelial, mesenchymal and musculature layers of the small and large intestine (semi-quantitative results in Supplementary Table 5, micrographs in Supplementary Fig. 1b and 1c, negative control in Supplementary Fig. 1d).

On the protein level, immunoreactivity of DKK1 was detected in various layers of the human small and large intestine, including the enteric nervous system (Fig. 1c-d, Supplementary Fig. 2). Our stainings confirmed no DKK1 expression in smooth muscle cells in the *Tunica muscularis* nor in the muscular *Tunica media* of arterioles within the intestinal wall (Supplementary Fig. 2d, arrow). Further, we found a homogeneous punctuated cytoplasmic DKK1 expression in some cells located at the crypt-bottom (Supplementary Fig. 2b small intestine and Supplementary Fig. 2c large intestine, arrow). In the enteric nervous system, we found intensively fluorescent punctae in the neuropil, suggesting a considerable DKK1-expression in neural processes within the ganglia (Fig. 1c-d, Supplementary Fig. 3a-l and Supplementary Fig. 2a for negative controls). However, the somata of enteric neurons exhibited only a weak homogeneous cytoplasmic DKK1-immunoreactivity, both in submucosal (Fig. 1c-i and d-i) and myenteric ganglia (Fig. 1c-ii and d-ii) of small and large intestinal samples (Fig. 1c-d, cutouts are optical sections, Supplementary Fig. 3a-l). Interestingly, we detected some staining for DKK1 in PGP9.5-co-labeled neurites outside the ganglia within the *Tunica muscularis* and in the fine neurite network surrounding the mucosal crypts (Supplementary Fig. 2d-e, arrowheads). Further, we found DKK1 co-localization with GFAP in the glial cells surrounding the mucosal crypts (i.e., type-III enteric glial cells, Supplementary Fig. 2f, arrowheads).

**DKK1 antagonizes canonical Wnt-signaling in murine enterospheres.** To evaluate the molecular function of DKK1, we carried out pharmacological stimulation studies in the in vitro model system of murine enterospheres. Therefore, we generated enterospheres isolated from the *Tunica muscularis* of postnatal mice. After culturing for 5 days in vitro, enterospheres were collected and the expression of DKK-ligands and -receptors was analysed using RT-PCR (Fig. 2a). In previous studies we already reported the expression of Wnt-signaling-inducing frizzled receptors[21]. We found a robust mRNA-expression of the ligands *Dkk1, 2* and *3* (but not *Dkk4*), the transmembrane-protein *Krm1* (but not *Krm2*), as well as the co-receptors *Lrp5* and *6* (Fig. 2b-c, Supplementary Fig. 4a). In addition, we verified the expression of *DKK1* and *KRM1* on human intestinal epithelial Caco-2 cells, as a positive control[33], (Supplementary Fig. 4a-b). Taken together, our in vitro model system is functionally equipped with the receptor repertoire to react upon an external DKK1-stimulus.

Moreover, we detected a strong mRNA expression of *Dkk1* in cultures of FACS-purified murine ENS cells (Supplementary Fig. 5a) as well as DKK1-immunoreactivity within proliferating human enterospheres. Interestingly, we found that DKK1 co-localized with P75⁺-somata (Supplementary Fig. 5b-i-ii, arrowheads) as well as P75⁺-fibers (Supplementary Fig. 5b-i-ii, arrows),

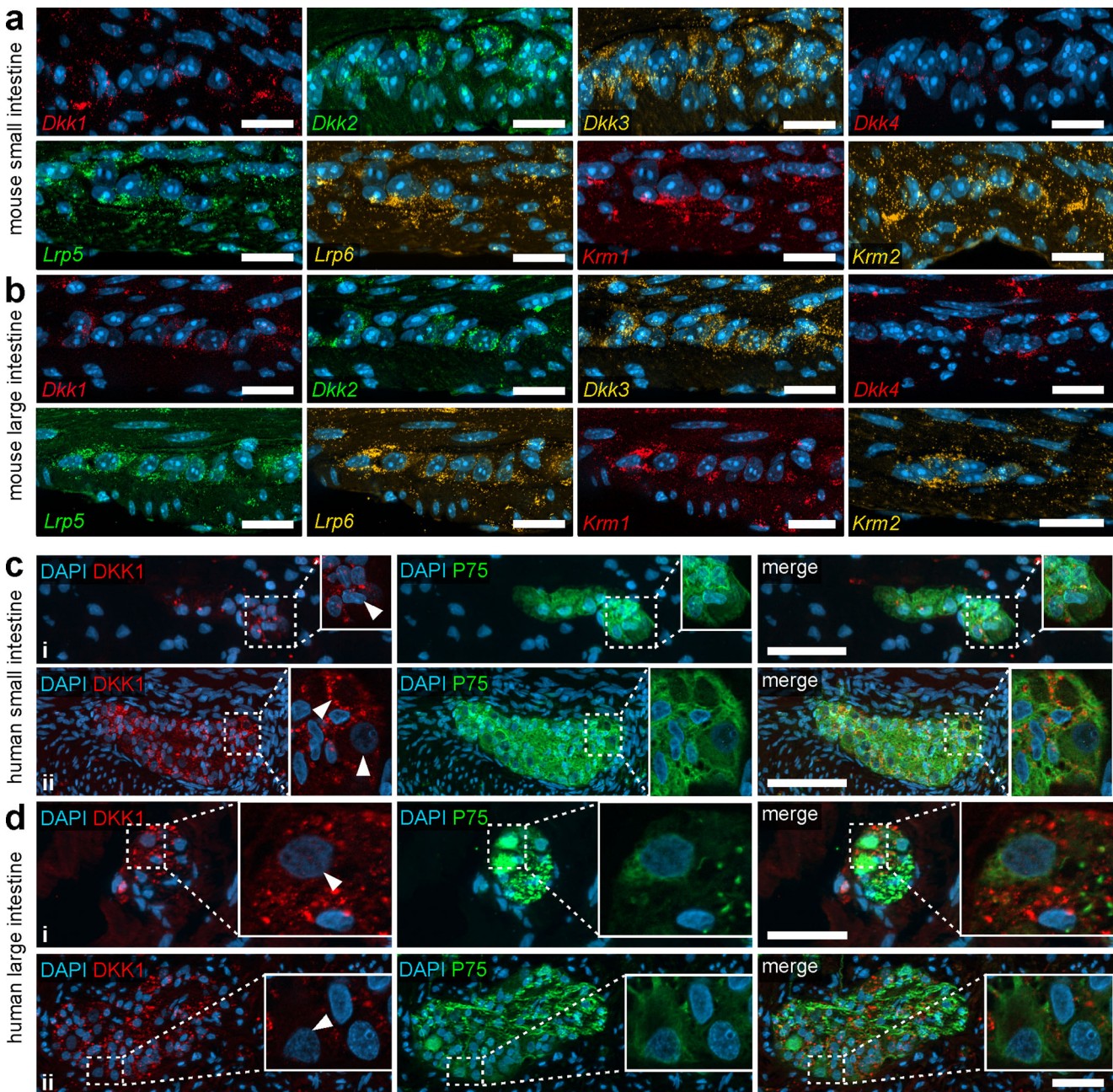

**Fig. 1 DKK1 is expressed in the murine and human ENS. a, b** In situ hybridization experiments showed *Dkk1, Dkk2, Dkk3, Dkk4, Krm1, Krm2, Lrp5* and *Lrp6* mRNAs detection (colors as indicated) and the nuclear marker DAPI (blue) within myenteric plexus of murine small (**a**) and large intestine (**b**) with different expression levels. **Scale bars: 20 μm. c, d** Immunofluorescence co-labeling studies with DKK1 (red), the neural crest marker P75 (green), and the nuclear marker DAPI (blue) indicate a strong DKK1-expression in the neuropil of submucosal (**c-i** and **d-i**) and myenteric (**c-ii** and **d-ii**) ganglia of human small and large intestine. However, little to no immunoreactivity was detected in neuronal somata (high-power magnifications are optical sections, arrow heads indicate enteric neurons). **Scale bars: 50 μm.**

(Supplementary Fig. 5biii for negative control). This strongly suggests that similar to our findings in vivo, DKK1 is expressed by neural cells of the human and murine ENS in vitro.

The functional modulation of the canonical Wnt-signaling pathway was analysed by quantification of active β-CATENIN levels by Western-Blot as well as of mRNA-expression levels of three well-known target genes *Axin2, Lef1* and *Lgr5* by qRT-PCR experiments[34–36]. To modulate the activation of the Wnt-signaling cascade, we used WNT3A and DKK1. The first one is a member of the Wnt protein family, that is used to trigger canonical Wnt-signaling activity, whereas the latter leads to

proteosomal degradation of β-CATENIN and downregulation of Wnt target genes[37] by Krm1/2-LRP5/6 receptor clearance from the cell membrane[38,39]. After the expansion of enterospheres for 5 days, cultures were either stimulated with DKK1 (500 ng/ml) or WNT3A (100 ng/ml) for one, two, and four hours for Western-Blot and six hours for qRT-PCR experiments. Western-Blot analysis demonstrated, that DKK1-stimulation resulted in a decrease of median active-β-CATENIN protein level after four hours (1 h: 0.44-fold ($P > 0.05$), 2 h: 1.07-fold ($P > 0.05$), 4 h: 0.57-fold ($P < 0.05$); $n = 3$; Fig. 2D-E, Supplementary Fig. 6a), whereas WNT3A-stimulation led to an increase in median active-β-

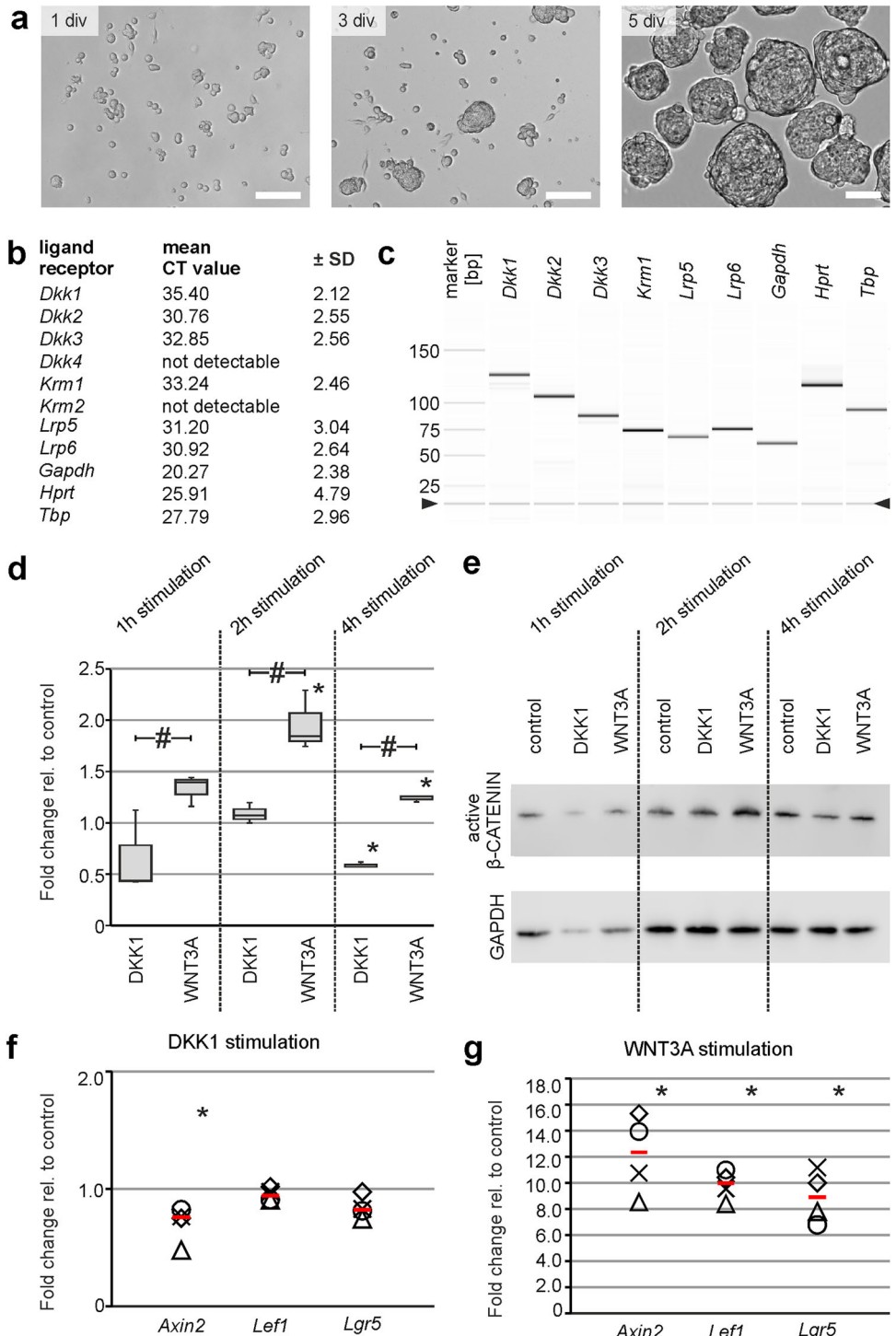

**Fig. 2 DKK1 antagonizes canonical Wnt-signaling in murine ENS-progenitors. a** Depicts enterospheres generated from isolated murine ENS-progenitors after 1, 3, and 5 days in-vitro (div). **Scale bars:** 1 and 3 div: **100 µm**, 5 div: **50 µm**. **b** Identification of DKK-ligands *Dkk1-Dkk3*, DKK-receptor *Krm1*, and the co-receptors *Lrp5* and *Lrp6* in enterospheres by qRT-PCR (CT-values expressed as mean ± SD) and in **c** by capillary electrophoresis (arrowhead indicates position of alignment marker). **d**, **e** Median active-β-CATENIN levels decreased after DKK1-stimulation within 4 hours, whereas WNT3A-stimulation led to an upregulation of median active-β-CATENIN levels relative to the untreated control group, quantified by Western-Blot experiments. The asterisk indicates significant differences compared to control group and the hash displays significant difference between groups (ANOVA on Ranks, Student-Newman-Keuls post hoc test; $n = 3$; 1 h: DKK1 vs. control: $P > 0.05$, WNT3A vs. control: $P > 0.050$, WNT3A vs. DKK1: $P < 0.050$; 2 h: DKK1 vs. control: $P > 0.050$, WNT3A vs. control: $P < 0.050$, WNT3A vs. DKK1: $P < 0.050$; 4 h: DKK1 vs. control: $P < 0.050$, WNT3A vs. control: $P < 0.050$, WNT3A vs. DKK1: $P < 0.050$).
**f** Median fold change of expression levels (red bar) of Wnt-target genes A*xin2*, *Lef1* and *Lgr5* decreased in DKK1-stimulated enterospheres while **g** WNT3A-stimulation led to an upregulation of Wnt-target genes, quantified by qRT-PCR. Data points are represented by different symbols. Asterisk indicates significant differences compared to the control group (Mann-Whitney Rank Sum Test; $n = 4$; DKK1-group: Axin2: $P = 0.029$, Lef1: $P = 0.343$, Lgr5: $P = 0.343$; WNT3A-group: Axin2: $P = 0.029$, Lef1: $P = 0.029$, Lgr5: $P = 0.029$).

CATENIN protein level (1 h: 1.40-fold ($P < 0.05$), 2 h: 1.85-fold ($P < 0.05$), 4 h: 1.26-fold ($P < 0.05$); $n = 3$; Fig. 2d-e, Supplementary Fig. 6a).

Under DKK1-stimulation the Wnt-target genes *Axin2* (median: 0.76-fold; $n = 4$; $P = 0.029$), *Lef1* (median: 0.94-fold; $n = 4$; $P = 0.343$) and *Lgr5* (median: 0.82-fold; $n = 4$; $P = 0.343$) were downregulated in comparison to untreated control cultures (Fig. 2f). In contrast, WNT3A-stimulation led to an upregulation of all three Wnt-target genes in comparison to control cultures: *Axin2* (median: 12.7-fold; $n = 4$; $P = 0.029$), *Lef1* (median: 10.3-fold; $n = 4$; $P = 0.029$) and *Lgr5* (median: 8.89-fold; $n = 4$; $P = 0.029$), (Fig. 2g). It is noteworthy that neither the pharmacological stimulation with either DKK1 or WNT3A (Supplementary Fig. 6b), nor the supplementation with EGF/FGF as well as single application of EGF or FGF (Supplementary Fig. 6c) altered the gene expression of three widely used housekeeping genes *Gapdh, Hprt* and *Tbp* in enterospheres. In addition, growth factor stimulation had no effect on *Dkk*1 expression as well (Supplementary Fig. 6c). Intriguingly, *Dkk*1 expression was significantly upregulated 3 hours after exogenous DKK1-stimulation, suggesting a contra-intuitive autoregulation process that demands future investigations (mean CT ± SD: 3 h: DKK1 vs. control: $P = 0.041$; 6 h: DKK1 vs. control: $P = 0.142$; 12 h: DKK1 vs. control: $P = 0.592$; $n = 3$; Supplementary Fig. 6d). To conclude, our data indicates that ENS-progenitors are functionally equipped with a suitable ligand and receptor combination to downregulate canonical Wnt-signaling cascade upon an external DKK1-stimulus.

**DKK1 has a pro-proliferative effect on murine P75$^+$ neural cells.** To evaluate the cell biological function of DKK1, ENS-progenitors derived from postnatal mice were expanded 5 days under proliferative conditions. After one day in vitro (1 div), cultures were stimulated with DKK1 (500 ng/ml), WNT3A (20 ng/ml) alone or in combination. Untreated cultures served as controls. At 5 div, the number and volume of proliferating enterospheres was analysed (Fig. 3a). All three experimental conditions led to an increase in the number (median: DKK1: 1.69-fold ($P < 0.05$); WNT3A: 1.90-fold ($P < 0.05$); Combination: 1.71-fold ($P < 0.05$); $n = 3$) and in the cumulative volume (mean ± SD: DKK1: 2.45 ± 0.89 ($P \leq 0.001$); WNT3A: 2.15 ± 0.37 ($P = 0.002$); Combination: 1.685 ± 0.45 ($P = 0.041$); $n = 3$) of enterospheres in comparison to the control group (Fig. 3b-d). This effect was reproducible in human enterosphere cultures: At 14 div, all three experimental conditions led to an increase in the number (mean fold change ±SD: DKK1: 1.76 ± 0.56 ($P = 0.020$); WNT3A: 2.17 ± 0.18 ($P = 0.002$); Combination: 1.96 ± 0.25 ($P = 0.006$); $n = 3$), as well as in the cumulative volume (mean fold change ±SD: DKK1: 3.11 ± 0.55 ($P = 0.002$); WNT3A: 2.68 ± 0.76 ($P = 0.006$); Combination: 3.92 ± 0.59 ($P \leq 0.001$); $n = 3$) of human enterospheres in comparison to the control group (Supplementary Fig. 7a-b).

This increase in number and volume could be due to changes in the proliferative capacity of ENS-progenitor cells. To address this, we used Ki67- and PCNA-immunohistochemistry on sections of proliferating murine enterospheres after 5 div stimulated with DKK1 (500 ng/ml). Untreated enterospheres served as controls. The percentage of Ki67$^+$DAPI$^+$ cells of all DAPI$^+$ cells significantly increased by 1.75-fold (mean ± SD: Control: 22.0% ± 0.32%; DKK1: 38.7% ± 2.34%; $n = 3$; $P \leq 0.001$), as well as the percentage of PCNA$^+$DAPI$^+$ cells of all DAPI$^+$ cells by 1.1-fold (mean ± SD: Control: 78.6% ± 2.51%; DKK1: 87.0% ± 2.45%; $n = 3$; $P = 0.010$) in DKK1-stimulated spheres compared to the unstimulated control (Fig. 3e-f).

To determine which cell population was proliferating, we carried out immunohistochemistry for P75 to stain for neural cells[40]. The percentage of P75$^+$DAPI$^+$ cells of all DAPI$^+$ cells significantly increased in DKK1-stimulated enterospheres by 1.87-fold (mean ± SD: Control: 23.3% ± 2.66%; DKK1: 43.6% ± 3.39%; $n = 3$; $P = 0.001$) compared to control (Fig. 3g-h). This was paralleled by a strong and significant increase in Ki67-colabelled P75$^+$DAPI$^+$ neural cells after a DKK1-stimulus compared to control by 1.95-fold (mean ± SD: Control: 28.6% ± 8.11%; DKK1: 55.7% ± 3.45%; $n = 3$; $P = 0.006$), Supplementary Fig. 8a-b. In summary, our data indicates that DKK1 has a pro-proliferative effect on P75$^+$ neural cells in vitro.

**DKK1 does not increase enteric neurogenesis neither in mice nor in humans.** In order to find out whether the observed pro-proliferative effect leads to an increase of differentiated enteric neurons, we carried out BrdU-incorporation assays on murine and human enterosphere cultures, prior to differentiation (Fig. 4a). Immunocytochemical analysis showed that, BrdU$^+$HuC/D$^+$ cells could be found in all experimental groups and suggested that these cells derived from ENS-progenitor cells that proliferated in vitro (Fig. 4b). However, DKK1-stimulation did not increase the mean number of HuC/D$^+$ cells, compared to the control group. Interestingly, the combination of DKK1 and WNT3A significantly increased the amount of HuC/D$^+$ cells by 1.45-fold which was similar to the WNT3A-stimulation alone by 1.46-fold (mean ± SD: Control: 2194 ± 265; DKK1: 2331 ± 141 ($P = 0.588$); WNT3A: 3208 ± 113 ($P = 0.003$), Combination: 3180 ± 500 ($P = 0.004$); $n = 3$; Fig. 4c). Since WNT3A has a higher affinity to LRP5/6 binding sites as DKK1[37], it is conceivable that the WNT3A effect overruled DKK1 signaling in the combinatory treatment of enteric progenitor cells.

As shown in Fig. 4d, human enterospheres were cultured under proliferating conditions for 14 div, whereby drugs were added directly after seeding. BrdU was added at 13 div. After cell expansion, cell cultures were differentiated for 1 week before HuC/D/BrdU immunolabeling. Again, we could detect BrdU co-labeled neurons in all groups (Fig. 4e). Furthermore, the number of neurons did not increase during in vitro culture after a DKK1-stimulus compared to untreated control. Additionally, the combination exhibited a comparable effect as the WNT3A-stimulation alone (mean fold change ±SD: DKK1: 0.99 ± 0.03 ($P = 0.931$); WNT3A: 1.69 ± 0.08 ($P \leq 0.001$); Combination: 1.99 ± 0.12 ($P \leq 0.001$); $n = 4$; Fig. 4f and Supplementary Table 6 for absolute numbers). To conclude, our data demonstrates that although DKK1-stimulation has a pro-proliferative effect on P75$^+$ neural cells in vitro, it does not induce enteric neurogenesis in mice or humans.

**DKK1 does not increase enteric gliogenesis neither in mice nor in humans.** Kunke et al. demonstrated, that DKK1-mediated downregulation of Wnt-signaling is required for postnatal onset of gliogenesis in mouse-derived postnatal forebrain neurospheres[28]. To analyze, if the pro-proliferative effect of DKK1 on P75$^+$ neural cells supports enteric gliogenesis at the expense of enteric neurogenesis, we quantified the number of SOX10$^+$, S100β$^+$ or GFAP$^+$ enteric glial cells. This set of glial markers was reported to cover the diversity of enteric glial cell populations[41]. In line with this report, we detected enteric glial cells positive for SOX10 and S100β (over 90% of all cells expressed at least one glial marker, Supplementary Fig. 9a-b), but also cells solely marked with S100β or SOX10. Since over 90 % of marked glial cells were SOX10$^+$S100β$^+$, we focused on the analysis of these cells.

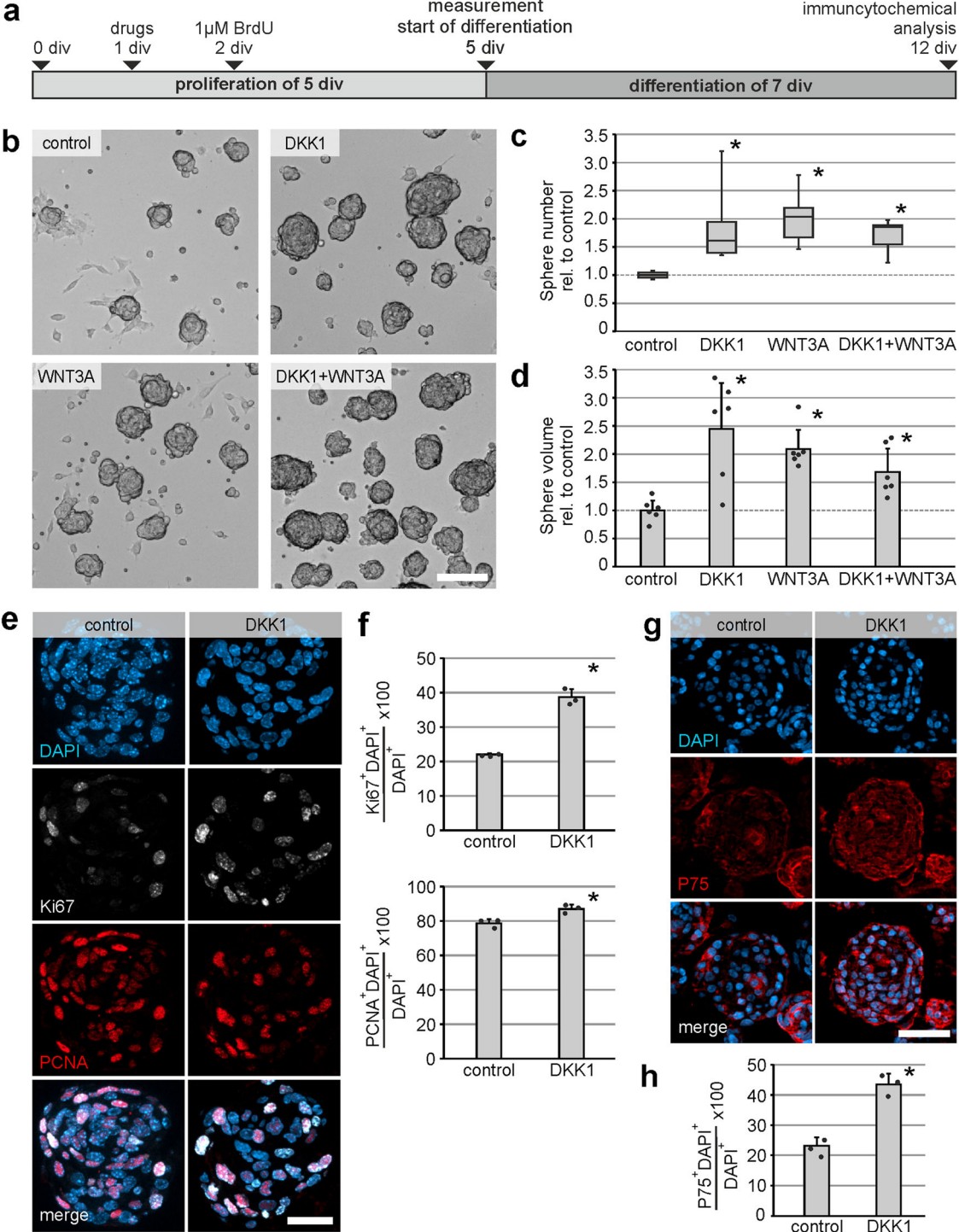

**Fig. 3 Pro-proliferative effect of DKK1 on murine P75$^+$ neural cells. a** Murine ENS-progenitors were cultured for 5 days in-vitro (5 div) under proliferative conditions. DKK1 and/or WNT3A was applied after 1 div. **b** Representative enterospheres cultures after 5 div, **Scale bar: 50 μm. c**, **d** All three experimental conditions increased the median number (boxplot: ANOVA on Ranks, Student-Newman-Keuls post hoc test; $n = 6$; DKK1 vs. control: $P < 0.050$, WNT3A vs. control: $P < 0.050$, DKK1 + WNT3A vs. control: $P < 0.050$) and the mean cumulative volume (barblot: ANOVA, Fisher LSD post hoc test; $n = 6$; DKK1 vs. control: $P \leq 0.001$, WNT3A vs. control: $P = 0.002$, DKK1 + WNT3A vs. control: $P = 0.041$) of enterospheres compared to control. Asterisk indicates significant differences to control, the dots represent individual data points. **e** Micrographs display immunfluoresence co-labeling studies with Ki67 (white) and PCNA (red) and the nuclear marker DAPI (blue), on paraffin-sections of 5-days-old enterospheres for the control and DKK1-treated group. **Scale bar: 10 μm. f** The barplots indicate the percentage of Ki67$^+$DAPI$^+$ and PCNA$^+$DAPI$^+$ cells (mean ± SD) for the control and DKK1-stimulated group. DKK1-stimulation increased the number of Ki67$^+$DAPI$^+$ and PCNA$^+$DAPI$^+$ cells. The asterisk indicates significant differences compared to control (Student t-test; Ki67: $n = 3$, $P \leq 0.001$; PCNA: $n = 3$, $P = 0.014$), the dots represent individual data points. **g** Micrographs show DAPI (blue) and P75 (red) stainings on paraffin-sections of 5-days-old enterospheres for the control and DKK1-treated group. **Scale bar: 40 μm. h** The barplot displays the percentage of P75$^+$DAPI$^+$ cells (mean ± SD), which significantly increased in DKK1-stimulated enterospheres. The asterisk indicates significant differences to control group (Student t-test; $n = 3$; $P = 0.001$), the dots represent individual data points.

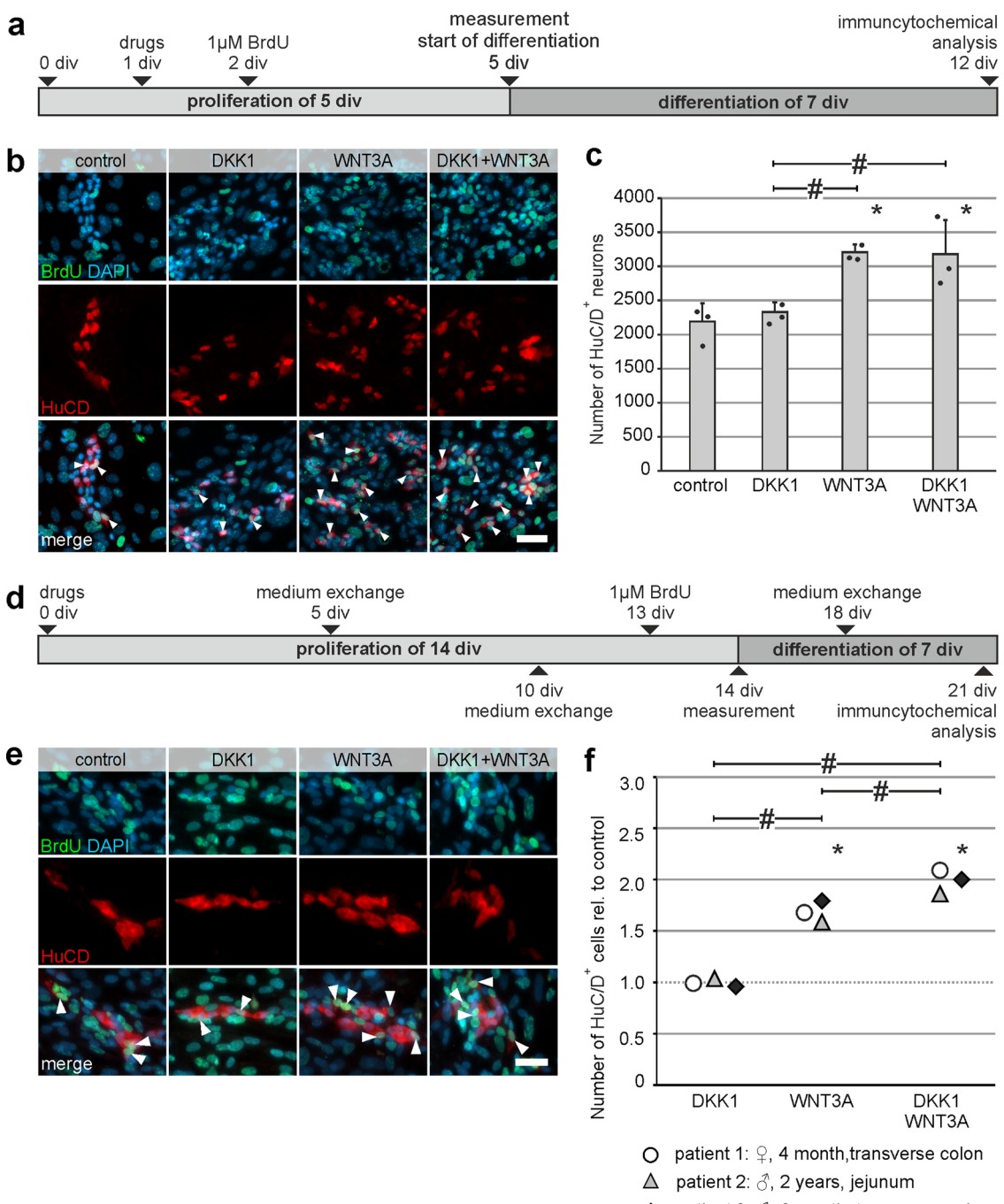

**Fig. 4 Pro-proliferative effect of DKK1 does not increase enteric neurogenesis in mice and human. a** Murine enterospheres were cultured for 5 days under proliferative conditions. DKK1 and/or WNT3A were added at 1 day and BrdU at 2 days in-vitro (1 and 2 div, respectively). After the proliferation phase, enterospheres were kept under differentiation conditions for one week. **b** Micrographs show immunofluorescence co-labeling studies in murine ENS-progenitors for BrdU (green) and the neuronal markers HuC/D (red) after 12 div. In all groups BrdU+HuC/D+ neurons were detected (arrow heads). **Scale bar: 50 μm. c** DKK1-stimulation did not increase the number of HuC/D+ cells compared to control (mean ± SD). Asterisk indicates significant differences compared to control and the hash depicts significant difference among the different groups (ANOVA, Fisher LSD post hoc test; $n = 3$; DKK1 vs. control: $P = 0.588$, WNT3A vs. control: $P = 0.003$, DKK1 + WNT3A vs. control: $P = 0.004$, WNT3A vs. DKK1: $P = 0.007$, DKK1 + WNT3A vs. DKK1: $P = 0.008$), the dots represent individual data points. **d** Human enterospheres were cultured for 14 days under proliferative conditions. DKK1 and/or WNT3A were added to the culture medium at the day of seeding and BrdU at 13 div. After the proliferation phase, enterospheres were kept under differentiation conditions for one week until 21 div. **e** BrdU-incorporation experiments in human ENS-progenitor cultures showed BrdU+HuC/D+ neurons in all groups (arrow heads). **Scale bar: 20 μm. f** Displays the quantification of the number of HuC/D+ neurons relative to control (mean ± SD). Data points for different patients are represented by different symbols. Number of neurons did not increase after DKK1-stimulation compared to untreated control. Asterisk indicates significant differences in comparison to the control group and hash depicts significant difference within groups (ANOVA, Fisher LSD post hoc test; $n = 3$; DKK1 vs. control: $P = 0.931$, WNT3A vs. control: $P \leq 0.001$, DKK1 + WNT3A vs. control: $P \leq 0.001$, WNT3A vs. DKK1: $P \leq 0.001$, DKK1 + WNT3A vs. WNT3A: $P = 0.002$, DKK1 + WNT3A vs. DKK1: $P \leq 0.001$).

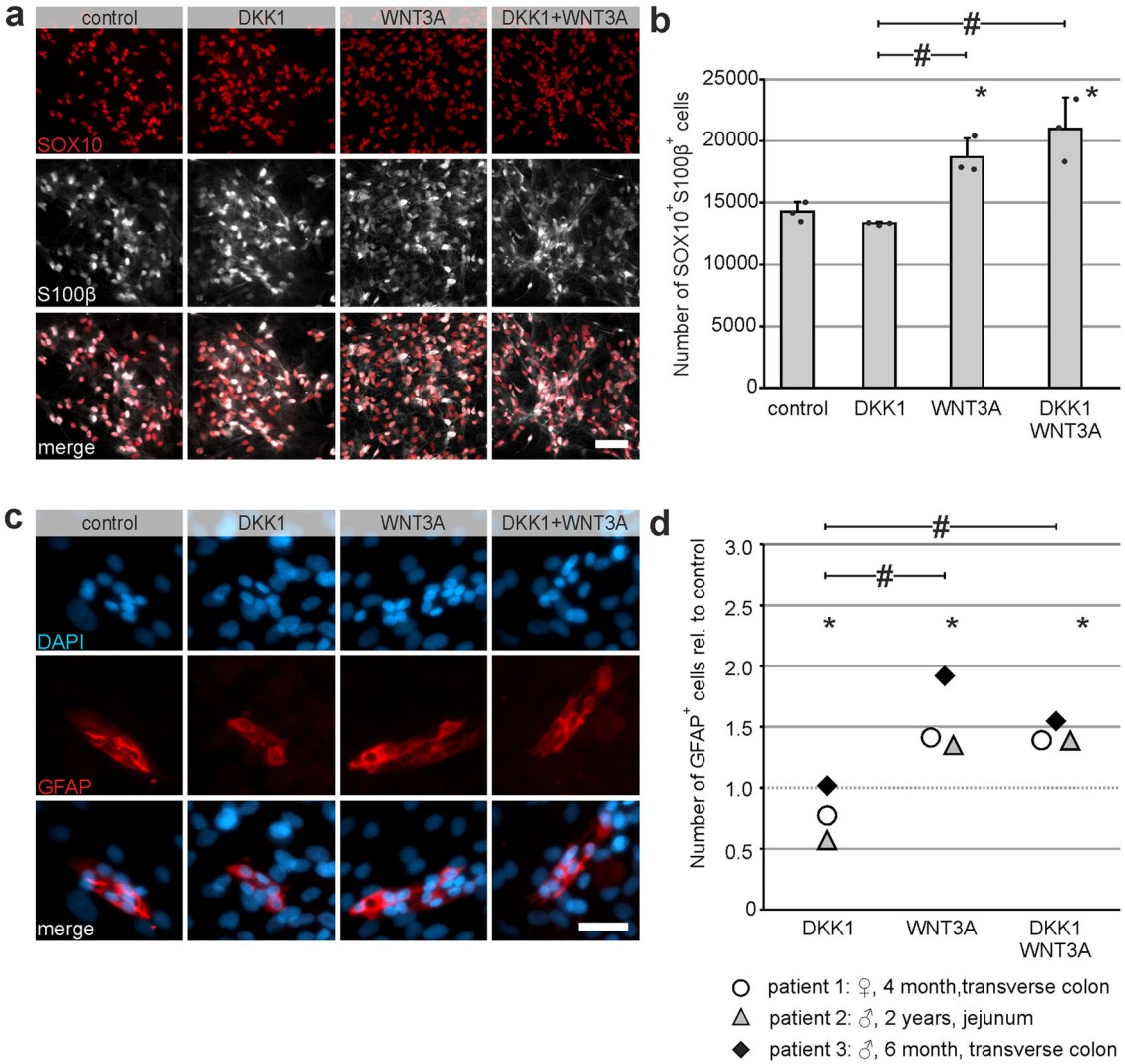

**Fig. 5 DKK1-stimulation does not enhance gliogenesis at the expense of neurogenesis in mice nor in humans.** For the quantification of enteric glia after DKK1-stimulation, murine enterospheres were cultured under proliferative conditions for 5 days in-vitro (5 div). DKK1 and/or WNT3A were added at 1 day in-vitro (1 div). Afterwards, enterospheres were cultured under differentiation conditions for 1 week. **a** Shows immunostaining for SOX10 (red) and S100beta (white). **Scale bar: 50 μm. b** The number of murine SOX10+S100β+ cells (mean ± SD) did not increase after DKK1-stimulation compared to control (barplot). Asterisk indicates significant differences compared to control and hash shows significant difference within groups (ANOVA, Fisher LSD post hoc test; $n = 3$; DKK1 vs. control: $P = 0.478$, WNT3A vs. control: $P = 0.008$, DKK1 + WNT3A vs. control: $P \leq 0.001$, WNT3A vs. DKK1: $P = 0.003$, DKK1 + WNT3A vs. DKK1: $P \leq 0.001$), the dots represent individual data points. **c** Human ENS-progenitors were cultured for 14 days under proliferative conditions. DKK1 and/or WNT3A were applied to the culture medium at the day of seeding. After the proliferation phase, enterospheres were kept under differentiation conditions for one week until 21 div. Micrographs depict co-staining for GFAP (red) and DAPI (blue). **Scale bar: 20 μm. d** The dot-plot shows the number of human GFAP+ enteric glial cells relative to control. Data points for different patients are represented by different symbols. DKK1-stimulation exhibited no significant effect compared to control. Additionally, the combinatory-treatment showed a comparable increase as the WNT3A-stimulation alone. Asterisk indicates significant differences compared to control and hash depicts significance among groups tested (ANOVA, Fisher LSD post hoc test; $n = 3$; DKK1 vs. control: $P \leq 0.001$, WNT3A vs. control: $P \leq 0.001$, DKK1 + WNT3A vs. control: $P \leq 0.001$, WNT3A vs. DKK1: $P \leq 0.001$, DKK1 + WNT3A vs. DKK1: $P \leq 0.001$).

DKK1-stimulation did not increase the mean number of murine SOX10+S100β+ cells compared to control group. The combinatory treatment significantly increased the number of SOX10+S100β+ cells by 1.47-fold and the WNT3A-stimulation alone by 1.31-fold (mean ± SD: Control: 14,251 ± 791; DKK1: 13316 ± 111 ($P = 0.478$); WNT3A: 18697 ± 1521 ($P = 0.008$), Combination: 20,995 ± 2551 ($P \leq 0.001$); $n = 3$; Fig. 5a-b, Supplementary Table 7 for absolute numbers).

Furthermore, we evaluated the total number of GFAP+ cells from human enterosphere cultures. Comparable to the results in mice, DKK1-stimulation did not significantly increase GFAP+ cells compared to the untreated control group. Additionally, the

combinatory treatment yielded a comparable significant effect as the WNT3A-stimulation alone (mean fold change ±SD: DKK1: 0.674 ± 0.13 ($P \leq 0.001$); WNT3A: 1.413 ± 0.07 ($P \leq 0.001$); Combination: 1.425 ± 0.06 ($P \leq 0.001$); $n = 4$; Fig. 5c-d and Supplementary Table 6 for absolute numbers). Taken together, this data demonstrates that although DKK1-stimulation has a pro-proliferative effect on P75+ neural cells in vitro, it does not induce enteric gliogenesis in murine nor human enterospheres.

**DKK1 induced cell-death in P75+ neural cells.** Several studies have shown, that DKK1 might be involved in cell-cycle control mediating apoptotic-mechanisms[42]. To further address the role of

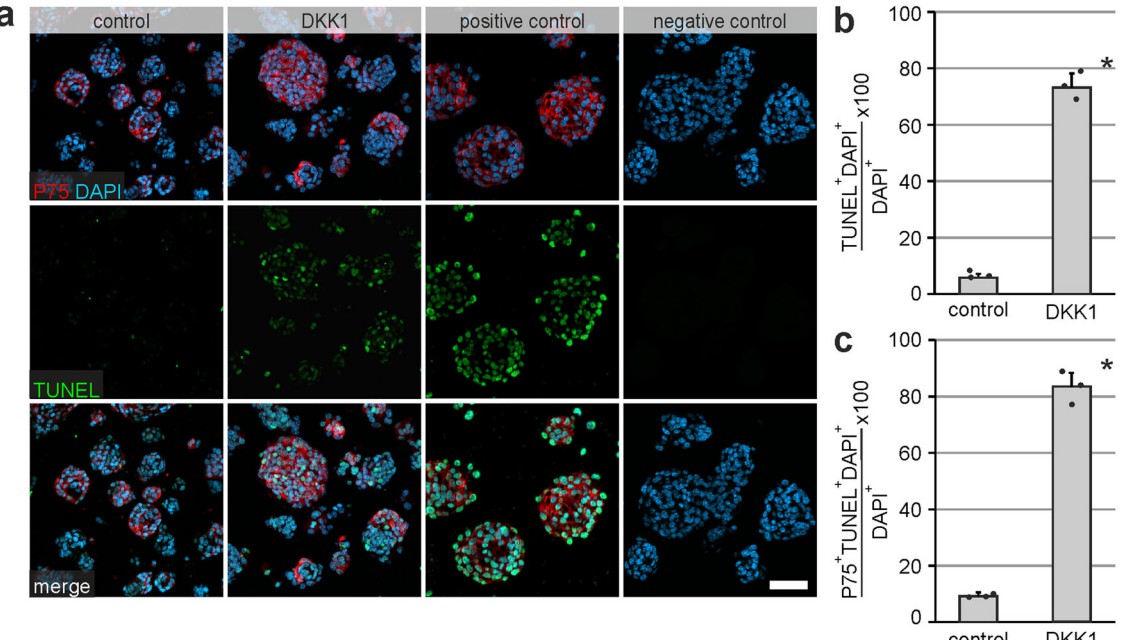

**Fig. 6 DKK1 induces cell-death in proliferating murine P75+ neural cells. a** Micrographs show immunofluorescence co-labeling studies with P75 (red), TUNEL (green) and DAPI (blue) performed on paraffin-sections of 5-days old enterospheres for DKK1-treated and the control group. Sections treated with DNAse were used as positive control. **Scale bar: 20 µm. b** The percentage of DAPI+TUNEL+ marked cells (mean ± SD) increased after DKK1-treatment compared to control. Asterisk indicates significant differences to the control (t-test; $n = 3$; $P \leq 0.001$), the dots represent individual data points. **c** Co-staining with P75 revealed an increase in the percentage of DAPI+TUNEL+P75+ cells (mean ± SD) in the DKK1-treated group compared to control. Asterisk indicates significant differences to control (t-test; $n = 3$; $P \leq 0.001$), the dots represent individual data points.

DKK1-mediated apoptosis, we first carried out terminal deoxynucleotidyl transferase–mediated deoxyuridine triphosphate nick-end labeling (TUNEL), that enables co-staining with P75 (Fig. 6a). We observed a significant increase in the percentage of DAPI+TUNEL+ cells of all DAPI+ cells by 13.1-fold in the DKK1-treated group compared to control (mean ± SD: Control: 5.67% ± 1.15%; DKK1: 73.0% ± 5.00%; $n = 3$; $P \leq 0.001$; Fig. 6b). Moreover, multiplexing with P75 immunostaining revealed an increase by 8.9-fold of DAPI+TUNEL+P75+ cells of all DAPI+ cells in the DKK1-treated group compared to the control group (mean ± SD: Control: 9.33% ± 0.57%; DKK1: 83.3% ± 6.03%; $n = 3$; $P = 0.001$; Fig. 6c). Next, to address this ambivalent function – increased sphere size versus increased cell death - we monitored enterospheres beyond 5 div until 8 div under proliferative conditions. We found that the number of cells with morphological signs of cell death increased steadily in culture after day 5 and was considerably higher than in control cultures (Supplementary Fig. 10). Yet, we still found that single spheroids increased in size, whereas other fell apart entirely. Thus, our data strongly indicates that DKK1 might function as a cell-death factor on ENS-progenitors in culture.

**DKK1-mediated cell-death acts via Caspase-3/7 and can be rescued.** Causeret and co-workers reported on the pro-apoptotic activity of the DKK1 co-receptor Krm1 arguably mediated by Caspase-3 activity[31]. Therefore, we stimulated proliferating murine enterospheres either with DKK1, the pan-Caspase-inhibitor zVAD-fmk or both, and carried out TUNEL-staining on enterospheres after five days in vitro. TUNEL-labeling of stimulated proliferating enterospheres revealed that DKK1-mediated cell-death could be rescued by the additional application of pan-Caspase-inhibitor zVAD-fmk (Fig. 7a). Furthermore, we evaluated DKK1-mediated Caspase-3/7 activity in real-time imaging using the NucView substrate. This substrate is cleaved by activated Caspase-3/7, resulting in an intracellular fluorescent signal. Similar

to the results from the TUNEL-assay, we found a strong and significant increase in the activity of Caspase-3/7 in response to a DKK1-stimulus (mean fold change ±SD on 2div: DKK1: 2.22 ± 0.16 ($P \leq 0.001$); zVAD-fmk: 0.458 ± 0.22 ($P = 0.001$); Combination: 0.914 ± 0.11 ($P = 0.455$); $n = 3$; Fig. 7b-c). This effect could be rescued by the co-application of the pan-Caspase-inhibitor zVAD-fmk. Moreover, we found that the detected Caspase-3/7-activation occurred rapidly after DKK1-application and declined considerably over the following days of observation (Supplementary Fig. 11a-b).

In addition, we evaluated the total number of enteric neurons and glial cells. We observed that DKK1+zVAD-fmk treatment significantly increased the number of differentiated HuC/D+ neurons by 1.76-fold (mean ± SD: Control: 1429 ± 267; DKK1: 1268 ± 321 ($P = 0.464$); zVAD-fmk: 957 ± 238 ($P = 0.053$); DKK1+zVAD-fmk: 2511 ± 169 ($P \leq 0.001$); $n = 3$; Fig. 7d) and the number of differentiated SOX10+S100β+ glial cells by 2.29-fold (mean ± SD: Control: 5152 ± 2506; DKK1: 5408 ± 2659 ($P = 0.876$); zVAD-fmk: 2047 ± 592 ($P = 0.087$); DKK1+zVAD-fmk: 11786 ± 1214 ($P = 0.003$); $n = 3$; Fig. 7e).

Interestingly, we found that the ratio of cells expressing SOX10 and/or S100β changed in response to pan-Caspase-inhibition, especially the number of SOX10+S100β+ glial cells decreased (Fig. 7e, Supplementary Fig. 11c-d, Supplementary Table 7 for absolute numbers). Moreover, it seems that enteric glial marker expression was more affected by pan-Caspase-inhibition than enteric neuron markers, hinting towards a previously unknown influence of Caspases on the differentiation of enteric glial cells, demanding future investigations. Taken together, these data strongly suggest, that DKK1 leads to a Caspase-dependent cell-death of ENS-progenitors in vitro and that this effect can be rescued by blocking Caspase-3/7-pathway.

**DKK1-mediated cell-death acts directly on ENS-progenitor cells.** Finally, to determine if the DKK1 has a direct effect on

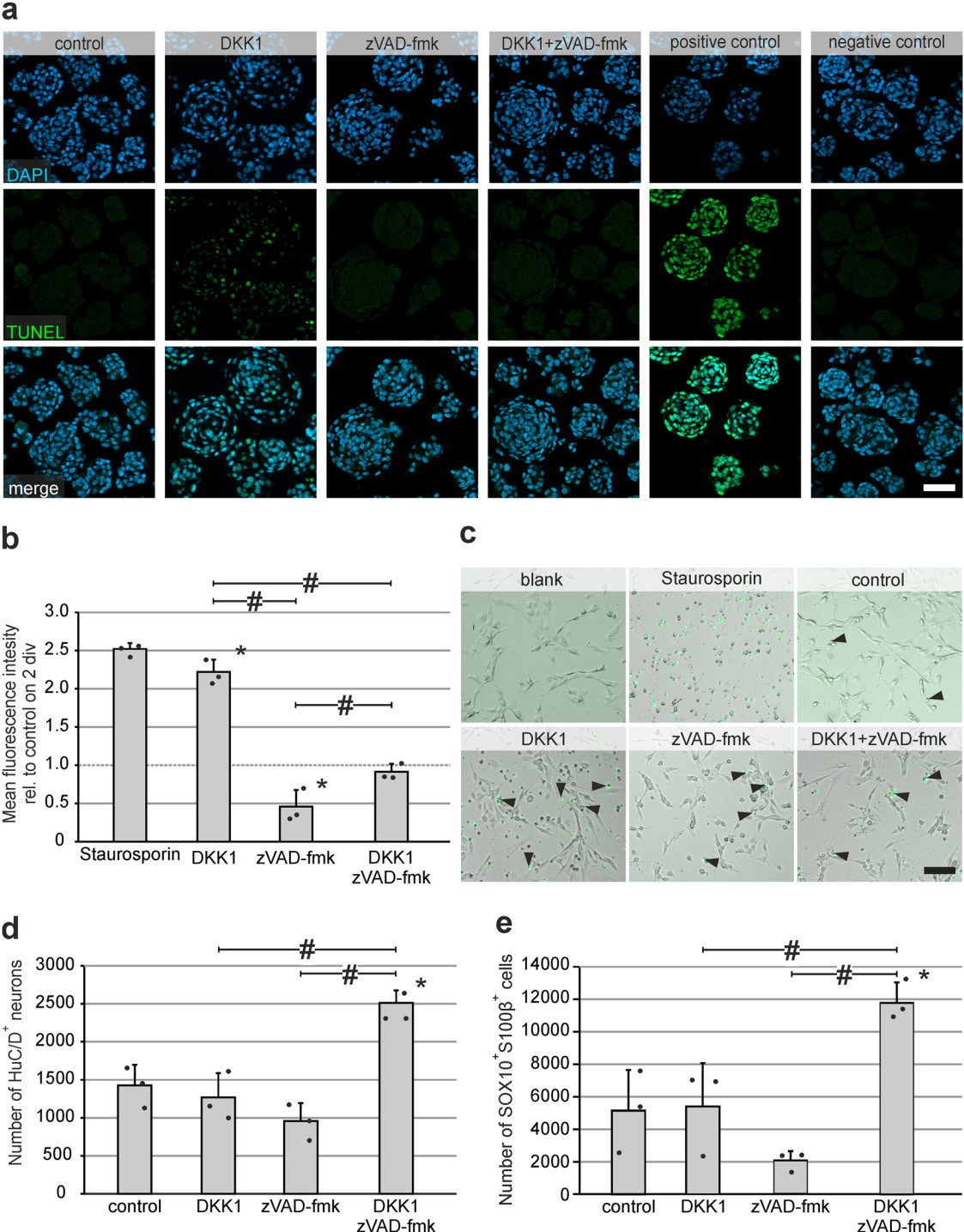

ENS-progenitors, we cultured FACS-purified neural crest−derived ENS cells from wnt1-tomato reporter mice under the same experimental conditions as outlined above (Fig. 8a). After differentiation, we could detect BrdU co-labeled neurons in all groups. However, the mean number of HuC/D$^+$ cells, compared to the control group did not increased after DKK1-stimulation. Moreover, the combination of DKK1 and WNT3A significantly increased the amount of HuC/D$^+$ cells by 1.81-fold which was similar to the WNT3A-stimulation alone by 1.46-fold (mean ± SD: Control: 20,166 ± 1031; DKK1: 20,777 ± 590; WNT3A: 29,439 ± 1202 ($P \leq 0.001$), Combination: 36,449 ± 2325 ($P \leq 0.001$); $n = 3$; Fig. 8b-c). In addition, we evaluated the total number of enteric neurons after DKK1+zVAD-fmk treatment.

Here, we again detected BrdU co-labeled neurons in all groups, but the number of differentiated HuC/D$^+$ neurons significantly increased by 1.68-fold (mean ± SD: Control: 20812 ± 1585; DKK1: 21,025 ± 1380; zVAD-fmk: 16,129 ± 1362; DKK1+zVAD-fmk: 35,124 ± 4344 ($P \leq 0.001$); $n = 3$; Fig. 8d-e). Together with our data from the proliferation assays (see above), these findings strongly support that the effect of DKK1 described in this work acts directly on ENS-progenitor cells in vitro.

## Discussion

In this study, we described the expression of DKK1 in the murine and human ENS and its influence on isolated ENS-progenitor

**Fig. 7 DKK1-induced cell-death in proliferating murine ENS-progenitors is rescued by pan-Caspase-inhibitor zVAD-fmk. a** Micrographs show TUNEL (green) and DAPI (blue) staining performed on paraffin-sections of 5-days old enterospheres after DKK1-, zVAD-fmk-, and DKK1+zVAD-fmk-treatment, as well as control. Sections treated with DNAse were used as positive control. No TUNEL-signal was detected in the zVAD-fmk-treated groups (with and without DKK1) suggesting rescue effects of DKK1-mediated DNA-fragmentation. **Scale bar: 40 μm. b** Shows the results of the in-vivo-Caspase-3/7-activity monitoring. Murine enterospheres proliferated for 5 days, whereby DKK1 and/or zVAD-fmk were added after 1 day in-vitro (1 div). Staurosporine-treatment served as positive control. The NucView 488 substrate was added to all experimental groups after 2 and cultures were analysed until 5 div. Cultures without substrate were used as blank. Co-application of the pan-Caspase-inhibitor zVAD-fmk blocked DKK1-mediated Caspase-3/7-activity (mean fluorescence intensity quantified in the barplot, mean ± SD). Asterisk indicates significant differences compared to control and the hash indicates significant difference between groups (ANOVA, Fisher LSD post hoc test; $n = 3$; DKK1 vs. control: $P \leq 0.001$, zVAD-fmk vs. control: $P = 0.001$, DKK1+zVAD-fmk vs. control: $P = 0.455$, zVAD-fmk vs. DKK1: $P \leq 0.001$, DKK1+zVAD-fmk vs zVAD-fmk: $P = 0.002$, DKK1+zVAD-fmk vs. DKK1: $P \leq 0.001$), the dots represent individual data points. **c** The micrographs show representative images of Caspase-3/7-activity in the different experimental groups (arrow head points to positive signal). **Scale bar: 100 μm. d** After differentiation, the DKK1-zVAD-fmk co-treatment increased the number of HuC/D+ neurons (mean ± SD). Asterisk indicates significant differences to control and the hash indicates significant difference between groups (ANOVA, Fisher LSD post-hoc test; $n = 3$; DKK1 vs. control: $P = 0.464$, zVAD-fmk vs. control: $P = 0.053$, DKK1+zVAD-fmk vs. control: $P \leq 0.001$, DKK1+zVAD-fmk vs. zVAD-fmk: $P \leq 0.001$, DKK1+zVAD-fmk vs. DKK1: $P \leq 0.001$), the dots represent individual data points. **e** The number of SOX10+S100β+ glial cells (mean ± SD) increased after DKK1-zVAD-fmk co-treatment. The asterisk indicates significant differences to control and the hash indicates significant difference between groups analyzed (ANOVA, Fisher LSD post-hoc test; $n = 3$; DKK1 vs. control: $P = 0.876$, zVAD-fmk vs. control: $P = 0.087$, DKK1+zVAD-fmk vs. control: $P = 0.003$, DKK1+zVAD-fmk vs. DKK1: $P = 0.004$, DKK1+zVAD-fmk vs. zVAD-fmk: $P \leq 0.001$), the dots represent individual data points.

cells in vitro. Our experiments showed that *Dkk1-4*-mRNA and corresponding receptor-mRNAs were present within murine ganglia, partly contradicting a previous report from 2005, which did not detect *Dkk1*- and *Dkk4*-mRNA by in situ hybridization[43]. However, since we verified DKK1 expression on protein-level in the human ENS, this discrepancy was possibly due to the novel, more sensitive detection system used in our study. Moreover, we detected DKK-mRNA and proteins at particularly high level within the neuropil within the ganglia. Thus, the involvement of neuronal and glial processes in intercellular DKK-signalling is very compelling and demands future investigations.

Still, although we found four different DKK-ligands in the ENS in vivo, we focused our functional investigation on DKK1, as for the others quite divergent functions were described depending on the cellular context. For instance, DKK2 is expressed during organogenesis in multiple organs with a partially overlapping expression pattern to DKK1. Therefore, it was suggested that it can act in an opposite fashion to DKK1, in the absence of its receptor Kremen2 and in the presence of further WNT-ligands[44,45]. For DKK3, on the other hand, it was shown to bind neither LRP6 nor Krm receptors, but instead seems to activate an array of different signaling cascades unlike the other DKK-ligands[38]. Finally, DKK4 appears to have a redundant function to DKK1[45], however, it only was weakly expressed on mRNA level in vivo and was not detectable in our ex vivo model system.

As mentioned above, neurogenesis in the mature ENS is rare[9], but can be triggered in vitro by EGF/FGF and further enhanced by the activation of e.g., Wnt/β-catenin-signaling[21], GDNF-signaling[46] or Serotonin-pathway[5]. Considering the expression of the Wnt-antagonist DKK1 in mice and humans, it is plausible that DKK1 operates as a negative regulator on the proliferation of ENS-progenitors in vivo. Thus, we tested this hypothesis by evaluating DKK1 on the in vitro model system of enterospheres. Beyond that, we conducted DKK1-stimulation experiments in spheroid cultures of Wnt1-Cre reporter mice. In these mice, neural cells express the red fluorescent protein tdTomato, which can be employed to generate purified neural cultures from *Tunica muscularis* using FACS sorting.

Our observation, that the stimulation with DKK1, had a pro-proliferative effect on P75+ neural cells in vitro was unexpected. Nevertheless, this observation is consistent with a previous report that highlighted DKK1 as a pro-proliferative factor in human adult mesenchymal stem cells allowing cell-cycle re-entry by

inhibiting Wnt/β-catenin-signaling[47]. Taking these results into consideration, Hartmann and Tabin proposed a model in which an increase in Wnt/β-catenin-signaling drove adult human mesenchymal stem cells out of the cell-cycle and pushed them towards differentiation and/or a postmitotic state[48].

However, data generated in neural tissues contradict the findings: Seib et al. reported that the inducible loss of DKK1 in quiescent neural progenitors in the subgranular zone of the hippocampus increased local Wnt-activity. This led to enhanced self-renewal and increased generation of amplifying neuronal progenitors and neuroblasts. Intriguingly, the increased pro-liferation rate of neural progenitors altered neither the number of newly-generated granule cells nor the proportion of newly-generated astrocytes[49]. In fact, the higher proportion of neuro-blasts underwent Caspase-3-mediated cell-death[49].

Similarly, the activation of Wnt/β-catenin-signaling promotes neurogenesis in ENS-progenitors in vitro[21]. Yet, the effect of DKK1 in the present study contradicts the findings by Seib et al. in being pro-proliferative on P75+ cells. However, this neural proliferation did not increase the yield of newly-generated neu-rons and glial cells, but instead was counterbalanced by an increased rate of Caspase-3/7-dependent cell-death.

In order to elucidate these ambivalent roles of DKK1, it is intriguing to investigate its crosstalk with epithelial growth factor (EGF) signaling. Niu and colleagues showed that EGF-receptor-(EGFR)-mediated-signaling induced DKK1-expression in hepa-tocellular carcinoma cells, thereby creating a negative feedback loop on sensitized cells[50]. They argue that, under physiological conditions and independent of EGFR-signaling, DKK1 negatively autoregulates its own expression. In fact, this feedback mechan-ism is known to be perturbed in human colon cancer, thereby promoting neoplastic progression[51]. Though there is little data available on the expression of growth factors in the ENS in vivo[52] enterosphere-formation and prolonged cell-expansion in vitro, relies on active growth-factor-signaling, and could not be trig-gered exclusively by stimulation of Wnt/β-catenin-signaling[21].

Our data showed, that the culture conditions under growth factor stimulation had no effect on endogenous DKK1 levels in vitro, concluding that the observed DKK1 effect on ENS-progenitors is not biased by the culture conditions per se. In addition, exogenous application of DKK1-protein upregulated *Dkk1* expression within the first three hours before cross-regula-tion, suggesting intracellular mechanism are present and immedi-ately active in vitro. In this cohesion, some reports suggested that

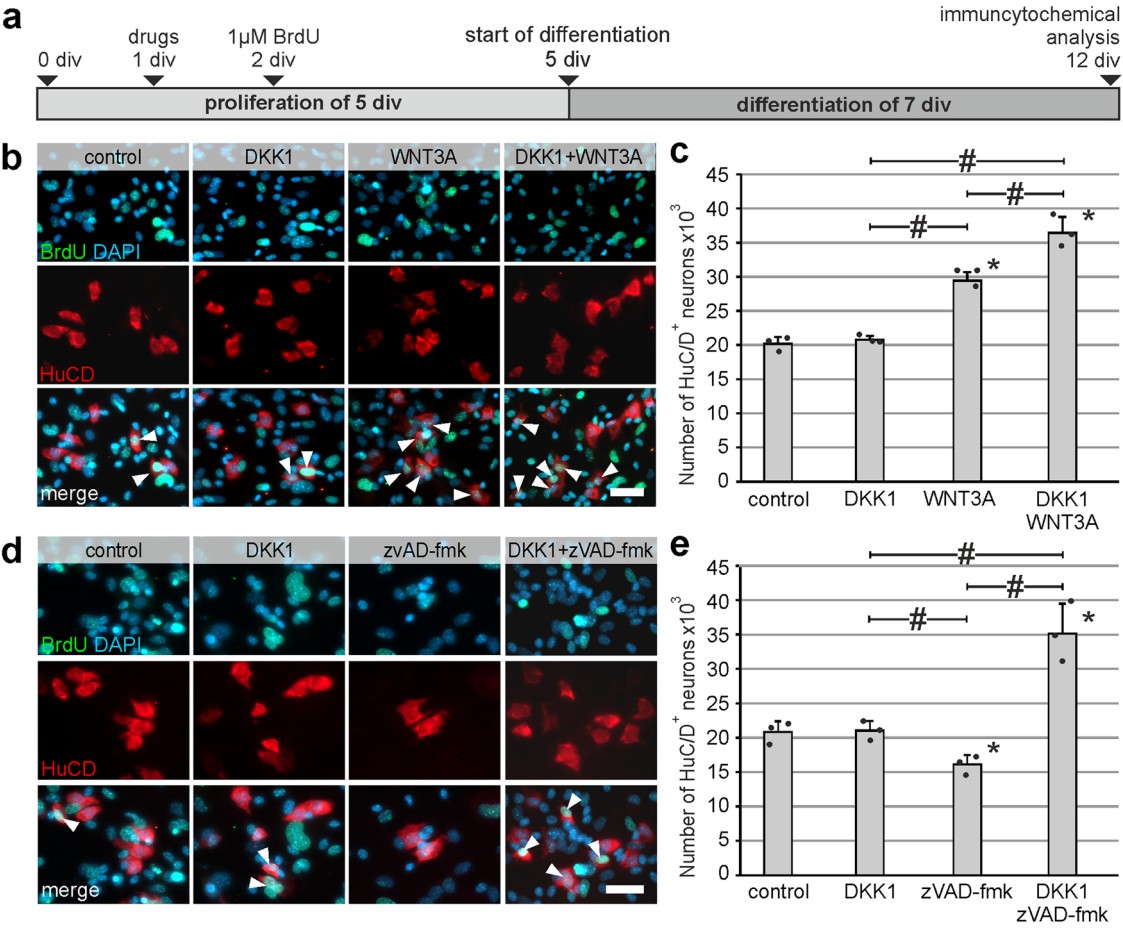

**Fig. 8 DKK1-effect and rescue from apoptosis is reproducible in purified ENS-progenitors isolated from wnt1-tomato reporter mice. a** FACS-purified ENS-progenitors were cultured for 5 days under proliferative conditions. DKK1 and/or WNT3A were added at 1 day and BrdU at 2 days in-vitro (1 and 2 div, respectively). After the proliferation phase, cells were kept under differentiation conditions for one week. **b** BrdU-incorporation experiments in purified murine ENS-progenitor cultures showed BrdU+ (green) and HuC/D+ (red) neurons in all groups (arrow heads). **Scale bar: 50 μm. c** Displays the quantification of the number of HuC/D+ neurons (mean ± SD). After differentiation, the number of neurons did not increase in response to DKK1-stimulation compared to untreated control. Asterisk indicates significant differences compared to control; hash indicates significant difference between groups (ANOVA, Fisher LSD post hoc test; $n = 3$; DKK1 vs. control: $P = 0.617$, WNT3A vs. control: $P \leq 0.001$, DKK1 + WNT3A vs. control: $P \leq 0.001$, WNT3A vs. DKK1: $P \leq 0.001$, DKK1 + WNT3A vs. WNT3A: $P \leq 0.001$, DKK1 + WNT3A vs. DKK1: $P \leq 0.001$), the dots represent individual data points. **d** Micrographs depict co-staining for HuC/D (red), BrdU (green) and DAPI (blue) in purified ENS-progenitors, arrowheads indicate BrdU+HuC/D+ cells. **Scale bar: 40 μm. e** After differentiation, the DKK1-zVAD-fmk co-treatment significantly increased the number of HuC/D+ neurons (mean ± SD). Asterisk indicates significant differences compared to control and the hash indicates significant difference between groups (ANOVA, Fisher LSD post hoc test; $n = 3$; DKK1 vs. control: $P = 0.920$, zVAD-fmk vs. control: $P = 0.051$, DKK1+zVAD-fmk vs. control: $P \leq 0.001$, DKK1 vs. zVAD-fmk: $P = 0.044$, DKK1+zVAD-fmk vs. zVAD-fmk: $P \leq 0.001$, DKK1+zVAD-fmk vs. DKK1: $P \leq 0.001$), the dots represent individual data points.

basal Wnt-activity is important to maintain cell-cycle-progression in dividing embryonic[53] and epithelial stem cells[54]. Moreover, DKK1-induced inhibition of Wnt/β-catenin-signaling interfered with cell-division by affecting microtubule-polymerization in mitotic cells[55], in turn favouring DNA-damage[56]. Taken together, the additional DKK1-stimulation and the resulting DKK1 over-expression could depress the obligatory Wnt-activity in entero-spheres within the first hours, thereby preventing further differentiation processes into more committed cell types and finally favouring cell-death, before cross-regulation takes place.

Moreover, there are different, partly contradictory, hypotheses on how DKK1 mediates cell-death on the molecular level. Causeret and co-workers reported on the pro-apoptotic activity of the DKK1-co-receptor Krm1 in a Wnt-independent manner[31]. These authors argue that Krm1 induces Caspase-3-activation, if DKK1-concentration falls below a threshold level. Here, we found *Krm1*-expression in vivo, as well as in proliferating enterospheres, making a Caspase-3-mediated cell-death of ENS cells via Krm1 conceivable.

Further, we showed that DKK1-mediated cell-death in enterospheres could be rescued by the co-application of the pan-Caspase inhibitor z-VAD-fmk. It should be mentioned, that Caspase-3-activity is not only implicated in cell-death, but also in other physiological processes[57]. In this context, Fernando et al., reported that neural stem cell differentiation is dependent on endogenous Caspase-3-activity and that its inhibition alters the expression of neuronal and glial proteins associated with neuro-sphere differentiation[58]. However, this process is not associated with neural cell-death, but led to a delayed differentiation process[58]. This mechanism could explain our results showing that zVAD-fmk treatment alone led to a cell-death-independent decrease in the number of enteric neurons and glial cells compared to control.

Trying to reconcile the pro-proliferative effect on entero-spheres on one side with the cell death-mediating function of DKK1 on the other side brings-up the aspect of cell death-induced proliferation. It was reported, that apoptotic mouse MEF

cells release mitogens, like WNT-ligands and growth factors like FGF, in a Caspase-3-dependent manner, thereby stimulating cell proliferation of stem- and progenitor cells and finally tissue regeneration[59]. In this occasion, we observed that in DKK1-stimulated enterospheres beyond 5 days of in vitro proliferation, single spheroids still increased in size, whereas others fell apart entirely with morphological signs of cell death, possibly suggesting that the pro-proliferative effect is stronger during the early proliferative culture phase, whereas cell death becomes stronger afterwards. Another explanatory approach to consider is the potential presence of different cell populations within the neural cell population, that have different susceptibility to DKK1-induced cell death. Thus, susceptible cells would die after a DKK1-stimulus, whereas non-susceptible populations might proliferate to make up for the loss. The analysis of variable receptor expression of different sub-populations and their contribution to intraganglionic cell-cell-signaling, however, is beyond the scope of this work and up to future studies.

We like to point out, that the data presented here does not assess the physiological function of DKK1 on postnatal ENS-progenitor cells in vivo as our functional results were gained from the analysis of in vitro models, only. Moreover, genetic approaches to differentiate the influence of endogenous vs. exogenous DKK1 was not part of the current investigation and demand future research. However, as Wnt-signaling serves as a multitude of vital functions in various tissues, manipulations, including for instance DKK1-knockouts are detrimental and not viable until the desired postnatal stage[60].

Another aspect we would like to highlight is, that DKK1 and the other members of the WNT-ligand-family are soluble factors[23], which often require or are modified by bridging molecules for receptor interaction, for instance the extracellular matrix. Back in 1999, Fedi and colleagues described that several Wnt signaling inhibitors like DKK1 bind to heperan[26]. As heperan is part of the sulfate chains of the matrix molecules heperan-sulfate-proteoglycans (HSPG), it is not surprising, that Caneparo et al. described DKK1-binding to the HSPG member Glypican 4 in zebrafish and Xenopus[61], which was later also outlined in human[62]. Since secretory HSPGs, such as collagen 18a and agrin were shown to have an influence on proper ENS development[15], our current findings on DKK1-signaling might hint towards underlying mechanisms of DKK-matrix interaction in the ontogenesis of the ganglionated plexus.

Moreover, the physical properties of the extracellular matrix were found to regulate endogenous DKK1 secretion. Barbolina and colleagues reported that matrix rigidity, but not mechanical strain, represses DKK1 expression and induces Wnt signaling – a mechanism important for cancer invasiveness[63]. As the mature ENS is constantly motile, ENS-progenitors may also respond to changes to this physical property of the surrounding extracellular matrix with a contra-proliferative behaviour in vivo. Thus, highlighting the important role of the microenvironment in the regulation of their cellular function not only during development[64] but also postnatally, that should be explored in the future.

Our study suggests, that ENS cells are functionally equipped with the suitable receptor/ligand repertoire in vivo, and that the enteric progenitor niche in vitro is tightly regulated by Wnt-signaling-agonists and -antagonist. However, as recently reviewed by Boesmann et al., enteric neurogenesis in the mature adult ENS and related neuronal cell-death is rare[8]. Thus, it is plausible that, although DKK-receptors/ligands are expressed in vivo in the postnatal murine and human ENS, additional cell-intrinsic and environmental factors are needed to stimulate proliferative potential of putative quiescent progenitors in vivo. Additionally, since enteric neuropathies may be associated with hyperganglionosis[65], the developmental importance of DKK1 for orchestrating an appropriate amount of proliferating enteric neural cells should be further investigated.

## Methods
For further details see Supplementary Material and Methods.

**Animals**. Animals were handled and kept in accordance with national guidelines, regulating the handling of animals for scientific purposes (Notification number AT 01/19 M), which conform international guidelines. For isolation of enteric neuronal progenitor cells, neonate (postnatal day 0–5) C57BL/6 J mice of both sexes were used. For in situ hybridization, small and large intestine samples from male C57BL/6 J mice (postnatal day 60) were used.

For fluorescence-activated cell sorting (FACS)-based isolation of ENS cells, we crossbred B6;129S6-Gt(ROSA)26Sortm9(CAG-tdTomato)Hze/J mice with B6.Cg-Tg(Wnt1-cre)2Sor/J mice. For simplicity, this crossbreed is termed wnt1-tomato in this article. For isolation of neural crest−derived ENS cells, neonate (postnatal day 0-5) wnt1-tomato mice of both sexes were used.

**Human Specimens**. Human gut samples were obtained from nine male and female patients aged between 3 months and 3 years who were operated due to imperforate anus, intestinal-obstruction syndrome, or short-gut syndrome (Supplementary Table 1). All samples were collected after approval by the local ethical committee (Project Nr. 652/2019BO2) and with the consent of the patients' parents according to the declaration of Helsinki.

**Cell culture of murine and human enteric neuronal progenitors**. The preparation of neonate murine and human enteric progenitor cells was carried out as described previously[21]. In brief, neonatal mice were decapitated and the *Tunica muscularis*, containing the myenteric plexus, was detached from the small intestine. After chopping the tissue, it was incubated in collagenase/dispase-solution and dissociated by three repetitive trituration and centrifugation steps. Cells were seeded in proliferation culture medium at a concentration of $5.0{\times}10^3$ cells/cm² or $2.0{\times}10^4$ cells/cm².

In some experiments, neural crest−derived cells were purified from wnt1-tomato reporter mice by FACS with a BD FACS Aria flow cytometer (BD Biosciences, Franklin Lakes, NJ). Forward-sideward scatter dot plots were used to exclude debris and cell aggregates. Endogenous tdTomato was excited by a 488-nm laser. Emission filter was 576/26 nm (see also Supplementary Fig. 12). Purified cells were seeded at a concentration of $1.0{\times}10^4$ cells/cm² on collagen-type-I coated cell culture plates.

For human specimens, the resectates were cut open along the longitudinal axis and rinsed twice with human preparation medium. The *Tunica serosa* and scar tissue were removed and the *Tunica muscularis* was peeled off the *Tela submucosa*. The tissue was chopped multiple times. The pieces were transferred to collagenase/dispase-solution dissolved in Hanks' balanced salt solution (HBSS) with $Ca^{2+}$/$Mg^{2+}$, dissociated by trituration, and washed in three centrifugation steps. Cell suspension was seeded at a concentration of $2.0{\times}10^4$ cells/cm². EGF and bFGF were added daily and culture medium was exchanged every 5 days.

Cultivation of murine and human ENS-progenitor cells under proliferation conditions resulted in 3-dimensional spheroids, termed *enterospheres*. For most of the experiments, we did not purify our cultures for neural cells, the resulting enterospheres also comprised non-neuronal cells of the *Tunica muscularis*, such as smooth muscle cells or fibroblasts. We consider this an advantage over purified neurospheres as it resembles the environment of the in vivo situation more accurately.

To activate or block the Wnt-signaling-pathway, we used WNT3A, DKK1 alone or in combination. Untreated cells served as control. For rescue experiments and Caspase-3/7 assay, cells were seeded at a concentration of $1.0{\times}10^5$ cells/cm² on collagen-type-I coated cell culture plates and stimulated either with DKK1 + DMSO, zVAD-fmk or their combination, DMSO-treated cells served as control.

Differentiation of enterospheres as well as for cultures seeded on collagen-type-I coated plates, was induced by removal of EGF and bFGF. To facilitate attachment of enterospheres, FCS was added to a concentration of 10 % (v/v) for 2 hours. All cultivation steps were conducted in a humidified incubator at 37 °C and 5 % $CO_2$.

**In situ hybridization**. We applied the RNAscope HiPlex Assay and HiPlex Assay v2 on 12 μm thick cryosections according to manufacturer's description with the following alterations. The target retrieval boiling time was adjusted to 5 min and incubation with Protease-III at 40 °C was adjusted to 5 minutes. HiPlex Target probes are specified in supplementary table 4.

**Immunostainings**. Deparaffinized sections were pre-treated with boiling citric acid monohydrate for three minutes and cooled down to room temperature. Cell cultures and tissue sections were pre-treated, followed by incubation of primary antibody overnight at 4 °C. For 5-bromo-20-deoxyuridine (BrdU) detection, cell

cultures were pre-treated with 2 N HCl at 37 °C for 30 min. Primary antibodies were detected using fluorescent secondary antibodies. Antibodies are listed in the Supplementary Table 2.

**Terminal deoxynucleotidyl transferase–mediated deoxyuridine triphosphate nick-end labeling assay**. We applied the Click-iT™ TUNEL Alexa Fluor™ 488 Imaging Assay Assay according to manufacturer's description to detect DNA damage associated with apoptosis on sections of 5-day-old enterospheres treated with or without DKK1 (Supplementary Table 2).

**Caspase-3/7-activity-assay**. To assess Caspase-3/7-activity we applied the Nuc-View 488 Caspase-3/7-substrate according to manufacturer's description.

**RNA isolation and RT-PCR**. Total RNA of enterospheres was isolated using the RNeasy Plus Mini Kit according to manufacturer's instructions. Reverse transcription was carried out with QuantiTect Reverse Transcription Kit. qPCR was performed using the StepOnePlus Real-Time PCR System the PerfeCTa qPCR ToughMix ROX according to manual. Supplementary Table 3 shows the primers for DKK-ligands and receptors, housekeeping genes, and Wnt-target genes.

**Western-blot analysis**. Proteins from enterospheres were isolated with RIPA buffer. 40 μg of each protein sample onto a 4-12 % Bolt Bis-Tris Plus Mini Gel and blotted onto a nitrocellulose membrane. Detection was carried out with the Pierce Fast Western-Blot Kit and the SuperSignal West Femto Substrate. The blots were scanned with the LICOR Odyssey XF Imaging System with an exposure time of 10 minutes each. The optical density was analysed using ImageJ (Supplementary Table 2).

**Statistics and reproducibility**. For all experiments, at least three independent experiments (independent preparations on different days) were carried out. Statistical analysis was performed using SigmaStat 3.5 software. Results were considered significant at $P \leq 0.05$. Statistical analysis for two groups was analyzed using Student's t-test or Mann-Whitney Rank Sum Test. Multiple groups were evaluated by either One-way Analysis of Variance (ANOVA) followed by Pairwise Multiple Comparison Procedure (Fisher LSD Method), or Kruskal-Wallis One-way Analysis of Variance on Ranks (ANOVA on Ranks) followed by all Pairwise Multiple Comparison Procedure (Student-Newman-Keuls Method). Bar blots are always shown as mean ± SD.

**Reporting summary**. Further information on research design is available in the Nature Portfolio Reporting Summary linked to this article.

## Data availability

We provided a *Supplementary data* file in Excel format with all numerical source data shown in the presented figures and charts (Supplementary Data 1). Additional information will be provided upon request.

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

## Acknowledgements

The project was supported by a grant from the German Research Foundation (DFG, Grant number: 438504601). We would like to thank Karin Seid and Melina Fischer for their technical assistance, Adrian Krestel for is help in the animal facility as well as Katharina Böhm, Lothar Just and Andreas Mack for their helpful comments on the manuscript.

## Author contributions

M.S.: acquisition of data; analysis and interpretation of data; drafting of the manuscript; critical revision of the manuscript for important intellectual content. S.S.: acquisition of data; critical revision of the manuscript for important intellectual content. B.H.: analysis and interpretation of data; critical revision of the manuscript for important intellectual content. P.H.N.: study concept and design; acquisition of data; analysis and interpretation of data; drafting of the manuscript; critical revision of the manuscript for important intellectual content; study supervision.

## Funding

## Competing interests

The authors declare no competing interests.
