## [Peer Review File · Communications Biology]

Reviewers' comments:

Reviewer #1 (Remarks to the Author):

In this manuscript Scharr and colleagues describe their experiments aimed at understanding the role of the Wnt antagonist Dickkopf1 (DKK1) in the regulation of enteric nervous system (ENS) progenitors. They combine in situ hybridization, immunofluorescence staining and an in vitro model systems (enterospheres) to investigate the expression of DKK1 related ligands and receptors in the gut wall, and their role in regulating the proliferation, survival and cell fate of ENS precursors. This is an interesting paper that touches upon an important topic in the field of ENS research.

My main concern is that what this manuscript brings in terms of advancement in our understanding on how DKK1 controls ENS progenitors in vivo, unfortunately, is rather limited. This is largely due to the fact that the functional experiments are performed in the enterosphere model only. The authors show that DKK-ligands and the related receptors are expressed by several other intestinal cell types next to those of the ENS. In this context, on several occasion throughout the manuscript the authors draw conclusions about ENS progenitors specifically, while in their culture model also other cell types are present, and likely express members of the DKK signaling machinery. Importantly, some of these other cells also proliferate (Figure 3), or are TUNEL+ (Figure 6) after DKK1 treatment. How does DKK1 expression in these cells compare to expression in the ENS progenitors present in the enterospheres? FISH or immunofluorescence experiments on the enterospheres would be very informative here. In addition, it is important to know whether DKK1 levels are affected by the culture conditions. Actually, in the discussion (P17, L350) the authors allude to such experiments. Of note, several of the concerns I raised here could be evaded by toning down the related conclusions, and by clearly stating that the observed effects occurred in vitro (e.g. also in the manuscript title).

Although the authors sort of touch upon this in the discussion section, it is not clear to me how they reconcile the effect of DKK1 stimulation on proliferation, which leads to an increase in the number and volume of enterospheres, with the effect on cell death, which would reduce enterosphere growth. Please explain.

As the title of this paper demonstrates, the manuscript focuses very much on neurogenesis. However, it turns out that the effects on gliogenesis in the enterosphere model are also quite significant, and possibly even more important (final results section). DKK1 signaling seems not to be specific for neurogenesis. This should be balanced better.

Some concepts described in the introduction are difficult to follow:

- P3, L58: The author's description of the neurogenic potential is confusing. By definition, ENS progenitors have neurogenic potential. Whether they engage their neurogenic potential indeed depends on several factors (such as the microenvironment as mentioned by the authors).
- P3, L61: What do the authors mean with 'regulatory cell-and molecular biological niche of ENS-progenitors'? The authors mention this requires further study. Is this niche something that is investigated in the current paper?
- P4, L73: the authors write that Dickkopf proteins are an evolutionary ancient gene family. To the best of my knowledge, every gene, if it were not a transgene, is the result of evolution. Maybe the authors mean that these genes are evolutionary conserved?
- P4, L85: Usually pro-neurogenic signals go at the expense of proliferative cues. Therefore, I find it surprising that the authors hypothesized that DKK1, in keeping with the fact that Wnt/beta-cat signaling is pro-neurogenic and DKK1 is a Wnt antagonist, would inhibit proliferation?

The authors should be congratulated for the nice and high quality fluorescence micrographs. Still, I have some questions. The PGP9.5 positive structures, by the authors defined as 'neuronal fiber bundles outside the ganglia' look a bit odd. Is there really such a high density of neuronal fibers running in bundles in the mucosal layers in human (Supp Fig 2D-E)? Why are there no fiber bundles

connecting to the ganglia visible? Also in Supp fig 2F: is the upper arrowhead really pointing to a GFAP+ DKK1+ structure?

In the results section, when reporting on multiple comparison tests, usually one P-value is mentioned, and significant differences between groups are depicted in the figures by an asterisk without exact P-values. However, for the Wnt target gene expression results shown in figure 2F (L157, P8), for each individual gene a P-value is mentioned. It would be better if this was consistent.

In figure 3C (upper panel) mean values are shown, at least that is what the Y-axis and legend say. In the results section median values are reported. Which one is correct?

For the experiments testing the effect of DKK1 on proliferation, WNT3a was used at 20ng/ml. In the other experiments 100ng/ml is used. Why is that?

Can Ki67 or PCNA labeling be combined with P75 staining to show that proliferation is indeed happening in ENS progenitors?

The final paragraphs in the results section (P14) describes changes in the Sox10/S100b ratio after caspase inhibition but it is not clear whether the differences between conditions are significantly different?

It is mentioned that the molecular-regulatory network of enteric neurogenesis is still in it's infancy, and a review article from 2016 is cited (ref 15). I am sure the authors are aware of the literature, some of which is quite extensive, that has become available since then. This statement should be amended.

There are several typos and errors that need to be fixed. P4, L82: suggests the. P6, L96: immunohistochemistry. P6, L116: Fig 1F is not there. Etc.

Reviewer #2 (Remarks to the Author):

This is an excellent manuscript that describes a role for Dickkopf1 (DKK1) in the development of the ENS. The authors find DKK1-4 transcripts and protein within the neuropil of each plexus of ENS ganglia. This observation is in contraction of a prior report, which the authors ascribe to the superior sensitivity of their methods. DKK1 is an antagonist of the action of Wnts. The authors find that DKK1 increases the proliferation of p75-immunoreactive neural precursor cells when added to enterospheres isolated from mature bowel. Because Wnt agonists promote neurogenesis, this action seems counterintuitive; however, the DKK1-induced proliferation is associated with extensive apoptosis. As a result, there is no net gain in numbers of viable neurons or glia; however, if caspases are inhibited with the pan-caspase inhibitor, zVAD-fmk, DKK1 treatment enhances both neurogenesis and gliogenesis. The authors acknowledge that their current work does not establish the physiologically relevant action of DKK1, but it does show that the ENS is endowed with both DKK1 expression and a relevant suite of receptors that enable DKK ligands potentially to interact with Wnts in controlling the homeostasis of the ENS.

The authors' story is compelling; however, there are details that require some discussion beyond that which is in the current manuscript. The figures do document DKK and its receptors in enteric ganglia, both at the transcript and protein levels; however, the figures do not make clear exactly where the ligands and receptors are. Additional clarity and localizing information would be helpful. In order to act, a ligand has to come into contact with its receptor. What triggers release of this particular ligand and how Dkk reaches its receptors are not apparent. The data were obtained with exogenous applications of ligand. This is artificial. What happens physiologically? How is endogenous Dkk activated? Actually, the use of enterospheres is also artificial and may bear little or no relationship to the in vivo situation. The authors, in their introduction appear to be very sensitive to the difference between in vitro and in vivo studies of enteric neurogenesis and speculate on potential roles of Wnt and DKK signaling; however, the experiments treat only a highly manipulated in vitro setup. Further

discussion, at least, of the proposed actions of endogenous DKK in vivo is warranted. The authors point out that DKK1-4 transcripts are expressed in the ENS but only DKK1 is studied as an exemplar of the group. That is okay but the authors should discuss why 4 different potential ligands are present in the ENS. Are they redundant or do they have different effects?

These are relatively small points and do not detract from the otherwise outstanding quality of the authors' work. The small points, however, are not trivial and they should be addressed.

We thank the reviewers for their useful comments on the manuscript. We made the requested changes to the manuscript and carried out several additional experiments to answer all of the reviewers' remarks. Please find our point-by-point responses (marked in blue) below.

Reviewer #1 (Remarks to the Author):

In this manuscript Scharr and colleagues describe their experiments aimed at understanding the role of the Wnt antagonist Dickkopf1 (DKK1) in the regulation of enteric nervous system (ENS) progenitors. They combine in situ hybridization, immunofluorescence staining and an in vitro model systems (enterospheres) to investigate the expression of DKK1 related ligands and receptors in the gut wall, and their role in regulating the proliferation, survival and cell fate of ENS precursors. This is an interesting paper that touches upon an important topic in the field of ENS research.

My main concern is that what this manuscript brings in terms of advancement in our understanding on how DKK1 controls ENS progenitors in vivo, unfortunately, is rather limited. This is largely due to the fact that the functional experiments are performed in the enterosphere model only. The authors show that DKK1-ligands and the related receptors are expressed by several other intestinal cell types next to those of the ENS. In this context, on several occasion throughout the manuscript the authors draw conclusions about ENS progenitors specifically, while in their culture model also other cell types are present, and likely express members of the DKK1 signaling machinery. Importantly, some of these other cells also proliferate (Figure 3), or are TUNEL+ (Figure 6) after DKK1 treatment. How does DKK1 expression in these cells compare to expression in the ENS progenitors present in the enterospheres?

We agree with reviewer 1, our data is limited to the *in vitro* situation and it is conceivable that other cell types contribute to the documented effects of DKK1 treatments. To tackle this, we conducted DKK1 stimulation experiments in spheroid cultures of Wnt1-Cre reporter mice. In these mice, neural cells express the red fluorescent protein tdTomato, which can be employed to generate purified neural cultures from *Tunica muscularis* using FACS sorting. Interestingly, we found that DKK1 had a comparable effect on the neuronal populations in purified neural cultures as in mixed enterosphere cultures (see added figure 8), strongly indicating that our findings indeed can be stated as a direct effect on ENS cells. Moreover, we changed the title and manuscript to clearly point out that these findings refer to the *in vitro* situation.

FISH or immunofluorescence experiments on the enterospheres would be very informative here.

We agree, this would add a new level of certainty to our findings. Thus, we used purified ENS cells of the Wnt1-Cre mouse model (see above) and carried out in situ hybridization for DKK1 after 5 days of culture under proliferative conditions. Indeed, these purified murine ENS cells exhibited an intense signal for *Dkk1* mRNA (supplementary figure 4A of the revised manuscript). Additionally, we carried out immunohistochemistry of sections of human enterospheres and were able to detect DKK1 expression especially in cells expressing the neural marker P75 (supplementary figure 4B of the revised manuscript).

In addition, it is important to know whether DKK1 levels are affected by the culture conditions. Actually, in the discussion (P17, L350) the authors allude to such experiments.

This point is well taken, but since the culture of enterospheres critically depends on the supplementation of growth factors, conceivable experimental changes to the culture conditions are limited. Thus, we carried out murine enterosphere cultures supplemented with EGF/bFGF (as usually in our laboratory), EGF alone, or bFGF alone. Interestingly, we did not detect any changes in the CT values of *Dkk1* and three established housekeeping genes in RT-qPCR analysis (supplementary figure

6c). Additionally, we carried out RT-qPCR experiments targeting *Dkk1* after a DKK1 treatment in murine enterospheres after 3, 6, and 12 h. Interestingly, we found a small, but significant upregulation of *Dkk1* mRNA in response to DKK1 treatment after 3 h, which, equilibrated to control levels at the 6 h timepoint (supplementary figure 6d). This indicated that while *Dkk1* expression is not altered by growth factor supplementation, there might be a somewhat contra-intuitive autoregulatory mechanism to increased DKK1 levels in the media. We changed the relevant paragraphs in the results and discussion sections accordingly.

Of note, several of the concerns I raised here could be evaded by toning down the related conclusions, and by clearly stating that the observed effects occurred in vitro (e.g. also in the manuscript title).

We agree, and thus change the title and parts of the manuscript accordingly.

Although the authors sort of touch upon this in the discussion section, it is not clear to me how they reconcile the effect of DKK1 stimulation on proliferation, which leads to an increase in the number and volume of enterospheres, with the effect on cell death, which would reduce enterosphere growth. Please explain.

This point is well taken and we struggled with this as well. We came up with two theories:

i) proliferation and cell death are run with different time scales, suggesting that the pro-proliferative effect is stronger during the observed proliferative culture phase (5 days in vitro), whereas cell death will become stronger at later stages. To tackle this, we kept enterospheres under proliferative conditions and DKK1 stimulation for more than 5 days. Indeed, we found that the number of cells with morphological signs of cell death increased steadily in culture after day 5 and was considerably higher than in control cultures (supplementary figure 10). Yet, we still found that single spheroids increased in size, whereas others fell apart entirely.

ii) different cell populations (maybe even within the neural cells) have varying susceptibility to DKK1-induced cell death. In this case, susceptible cells would die after a DKK1 stimulus, whereas non-susceptible populations might proliferate to make up for the loss. It is however conceivable that these populations are not strictly separable, but make up a spectrum, suggesting that modifier signals could regulate the DKK1 response. This question could be tackled by single cells analyses, however, this would be beyond the scope of this work. Therefore, we added a paragraph in the discussion section taking the described options into account.

As the title of this paper demonstrates, the manuscript focuses very much on neurogenesis. However, it turns out that the effects on gliogenesis in the enterosphere model are also quite significant, and possibly even more important (final results section). DKK1 signaling seems not to be specific for neurogenesis. This should be balanced better.

The changes to the title and manuscript described above took account of this point.

Some concepts described in the introduction are difficult to follow:

P3, L58: The author's description of the neurogenic potential is confusing. By definition, ENS progenitors have neurogenic potential. Whether they engage their neurogenic potential indeed depends on several factors (such as the microenvironment as mentioned by the authors).

P3, L61: What do the authors mean with 'regulatory cell-and molecular biological niche of ENS-progenitors'? The authors mention this requires further study. Is this niche something that is investigated in the current paper?

For better understanding we rephrased the “neurogenic potential” to “proliferative potential” in the introduction text. As already mentioned, whether a progenitor cell engages its proliferative potential into neurogenic or gliogenic direction depends on factors like morphogens, which are present in the surrounding extracellular matrix, thus forming a niche. These morphogens – like DKK1 and many others – have a regulatory influence by switching ON/OFF signaling pathways that are proliferative in general or pro-neurogenic or gliogenic (i.e., ‘regulatory cell- and molecular biological niche of ENS-progenitors’). We do not address the niche in vivo – as interventions on WNT signaling per se are quite detrimental for the corresponding organism. However, enterospheres are one possible in vitro model system to study for instance morphogens and their impact on proliferation/differentiation of ENS-progenitors.

We added these explanations to the respective sections in the text.

P4, L73: the authors write that Dickkopf proteins are an evolutionary ancient gene family. To the best of my knowledge, every gene, if it were not a transgene, is the result of evolution. Maybe the authors mean that these genes are evolutionary conserved?

Correct, we were referring to evolutionary conserved. We changed this sentence accordingly.

P4, L85: Usually pro-neurogenic signals go at the expense of proliferative cues. Therefore, I find it surprising that the authors hypothesized that DKK1, in keeping with the fact that Wnt/beta-cat signaling is pro-neurogenic and DKK1 is a Wnt antagonist, would inhibit proliferation?

As found in previous reports by our group and others, Wnt/b-cat signaling increases the proliferative rate of ENS progenitors, eventually leading to a higher number of newly generated neurons in vitro. Therefore, an anti-proliferative effect of DKK1 is primarily conceivable.

Yet, it is plausible that this effect will decrease in very long culturing periods, once all progenitor cells eventually left the proliferative state and committed to postmitotic fates. Also, one could speculate that a trade-off of proliferation vs. neurogenesis plays a more substantial role if the number of available progenitor cells is very limited (as possibly within a single ganglion).

The authors should be congratulated for the nice and high-quality fluorescence micrographs. Still, I have some questions. The PGP9.5 positive structures, by the authors defined as ‘neuronal fiber bundles outside the ganglia’ look a bit odd. Is there really such a high density of neuronal fibers running in bundles in the mucosal layers in human (Supp Fig 2D-E)? Why are there no fiber bundles connecting to the ganglia visible?

Indeed, the density of neuronal processes is very high in the lamina propria mucosae (see also Rao & Gershon 2016; <https://doi.org/10.1038/nrgastro.2016.107>). Although these authors used murine tissue, the resulting image looks very similar (see figure 2c in Rao & Gershon 2016). Since some of the detected structures are thicker than one would expect from a single neurite, we described them as neuronal fiber bundles. However, as our microscopic approach does not offer the resolution to distinguish between very small bundles and single axons, we changed the wording to “neurites” in the revised manuscript.

There mainly technical reasons, why interconnective strands cannot be appreciated very well in the images: We were using paraffin sections with a 5 µm thickness, while the submucosal plexus is rather large-meshed. Thus, hitting an entire interconnective bundle longitudinally, that connects two ganglia, that are within the same 5 µm plane is highly unlikely. However, as a matter of fact, we can see structures that likely resemble interconnective strands: (1) both submucosal ganglia in suppl. Fig 2D exhibit outgoing fiber bundles on their apical-lateral edges, (2) two smaller structures on the left side

of the same image likely resemble transversal sections of fiber strands. Therefore, we do not fully agree with the statement of reviewer 1 that no fiber bundles connecting ganglia were visible.

Also, in Supp fig 2F: is the upper arrowhead really pointing to a GFAP+ DKK1+ structure?

We changed this in the figure accordingly.

In the results section, when reporting on multiple comparison tests, usually one P-value is mentioned, and significant differences between groups are depicted in the figures by an asterisk without exact P-values. However, for the Wnt target gene expression results shown in figure 2F (L157, P8), for each individual gene a P-value is mentioned. It would be better if this was consistent.

We changed the manuscript accordingly.

In figure 3C (upper panel) mean values are shown, at least that is what the Y-axis and legend say. In the results section median values are reported. Which one is correct?

The data was not normally distributed; thus, the median is correct. We changed the Y-axis and legend accordingly.

For the experiments testing the effect of DKK1 on proliferation, WNT3a was used at 20ng/ml. In the other experiments 100ng/ml is used. Why is that?

Wnt3a concentrations of 100 ng/ml are relatively high and might reach a ceiling level of our readout (i.e., proliferation assays, neuron count). Since Wnt has a higher affinity to the relevant LRP5/6 receptors than DKK1 (see also Ahn et al. 2011), we chose a lower concentration in the current approach. We added this explanation to the result part.

Can Ki67 or PCNA labeling be combined with P75 staining to show that proliferation is indeed happening in ENS progenitors?

Yes, this can be done. Thus, we combined Ki67 with P75 staining in proliferating enterospheres validating our statement that proliferation is taking place in neural cells (supplementary figure 8). Moreover, our additional experiments using purified ENS cells from Wnt1-Cre reporter mice substantiate this (see above).

The final paragraphs in the results section (P14) describes changes in the Sox10/S100b ratio after caspase inhibition but it is not clear whether the differences between conditions are significantly different?

We added all p-values to the figure. However, we want to point out that despite a mathematical significance, the effect sizes are relatively low (<5% of cells) and we are rather skeptical if one should deduct a biological relevance from these findings.

It is mentioned that the molecular-regulatory network of enteric neurogenesis is still in its infancy, and a review article from 2016 is cited (ref 15). I am sure the authors are aware of the literature, some of which is quite extensive, that has become available since then. This statement should be amended.

We rewrote this paragraph and omitted the statement.

There are several typos and errors that need to be fixed. P4, L82: suggests the. P6, L96: immunohistochemistry. P6, L116: Fig 1F is not there. Etc.

We fixed the typos.

Reviewer #2 (Remarks to the Author):

This is an excellent manuscript that describes a role for Dickkopf1 (DKK1) in the development of the ENS. The authors find DKK1-4 transcripts and protein within the neuropil of each plexus of ENS ganglia. This observation is in contraction of a prior report, which the authors ascribe to the superior sensitivity of their methods. DKK1 is an antagonist of the action of Wnts. The authors find that DKK1 increases the proliferation of p75-immunoreactive neural precursor cells when added to enterospheres isolated from mature bowel. Because Wnt agonists promote neurogenesis, this action seems counterintuitive; however, the DKK1-induced proliferation is associated with extensive apoptosis. As a result, there is no net gain in numbers of viable neurons or glia; however, if caspases are inhibited with the pan-caspase inhibitor, zVAD-fmk, DKK1 treatment enhances both neurogenesis and gliogenesis. The authors acknowledge that their current work does not establish the physiologically relevant action of DKK1, but it does show that the ENS is endowed with both DKK1 expression and a relevant suite of receptors that enable DKK ligands potentially to interact with Wnts in controlling the homeostasis of the ENS.

The authors' story is compelling; however, there are details that require some discussion beyond that which is in the current manuscript. The figures do document DKK and its receptors in enteric ganglia, both at the transcript and protein levels; however, the figures do not make clear exactly where the ligands and receptors are. Additional clarity and localizing information would be helpful.

We agree with reviewer 2, that linking DKK1 expression to glial or neuronal processes (or even neuronal subtypes) within the neuropil would be helpful and highly interesting. However, our light microscopic approach does not offer sufficient resolution to make these statements. A conceivable option would be to use super-resolution microscopy for this purpose that we currently, however, do not have access to. We thus added a statement in the discussion section suggesting future investigations in this direction.

In order to act, a ligand has to come into contact with its receptor. What triggers release of this particular ligand and how Dkk reaches its receptors are not apparent. The data were obtained with exogenous applications of ligand. This is artificial. What happens physiologically? How is endogenous Dkk activated?

We are aware that our data refer to in vitro situations. We therefore changed the title and parts of the manuscript to make this issue more obvious to the reader. Moreover, we added information on endogenous Dkk activation to the discussion section.

Actually, the use of enterospheres is also artificial and may bear little or no relationship to the in vivo situation. The authors, in their introduction appear to be very sensitive to the difference between in vitro and in vivo studies of enteric neurogenesis and speculate on potential roles of Wnt and DKK signaling; however, the experiments treat only a highly manipulated in vitro setup. Further discussion, at least, of the proposed actions of endogenous DKK in vivo is warranted.

This is correct. Unfortunately, Wnt-signaling manipulations are rather tricky in the living animal as it serves a multitude of vital functions in various tissues. Nevertheless, we added a paragraph to the discussion section to speculate on DKK function in the living animal and in the ENS in particular.

The authors point out that DKK1-4 transcripts are expressed in the ENS but only DKK1 is studied as an exemplar of the group. That is okay but the authors should discuss why 4 different potential ligands are present in the ENS. Are they redundant or do they have different effects?

We added relevant information to the other potential ligands to the discussion part and why we focused on DKK1 in our study.

These are relatively small points and do not detract from the otherwise outstanding quality of the authors' work. The small points, however, are not trivial and they should be addressed.

REVIEWERS' COMMENTS:

Reviewer #1 (Remarks to the Author):

The authors have sufficiently addressed my questions and remarks. This is a nice piece of work that will be valuable to the field.

Reviewer #2 (Remarks to the Author):

The authors have done a nice job in revising their manuscript. The model, however, remains a model. The functional studies are essentially limited to neurospheres and there is no in vivo data backing up those experiments. There is also no attempt to determine whether the effects of exogenous administration of DJJ1 are at all like those of endogenous release.

Still, the authors have labored hard on this paper and the observations are of interest. It should be sufficient for the authors to make the limitations of their investigation more apparent. The discussion of the potential effect of the extracellular matrix on the action of DKK1 is interesting but not a substitute.

Reviewer #1 (Remarks to the Author):

The authors have sufficiently addressed my questions and remarks. This is a nice piece of work that will be valuable to the field.

We thank the reviewer for the helpful comments and time.

Reviewer #2 (Remarks to the Author):

The authors have done a nice job in revising their manuscript. The model, however, remains a model. The functional studies are essentially limited to neurospheres and there is no *in vivo* data backing up those experiments. There is also no attempt to determine whether the effects of exogenous administration of DJJ1 are at all like those of endogenous release.

Still, the authors have labored hard on this paper and the observations are of interest. It should be sufficient for the authors to make the limitations of their investigation more apparent. The discussion of the potential effect of the extracellular matrix on the action of DKK1 is interesting but not a substitute.

We thank the reviewer for the helpful comments and time.

We have added another paragraph pointing out the limitations of our study, specifically addressing the shortcomings that reviewer 2 points out (i.e., *in vitro* model and endogenous vs. exogenous DKK1 effects). Thus, we included the following statement in the discussion section:

“We like to point out, that the data presented here does not assess the physiological function of DKK1 on postnatal ENS-progenitor cells *in vivo* as our functional results were gained from the analysis of *in vitro* models, only. Moreover, genetic approaches to differentiate the influence of endogenous vs. exogenous DKK1 was not part of the current investigation and demand future research.”